# The evolution of parental care diversity in amphibians

Andrew I. Furness [1,2]* & Isabella Capellini [1,2,3]*

Parental care is extremely diverse across species, ranging from simple behaviours to complex adaptations, varying in duration and in which sex cares. Surprisingly, we know little about how such diversity has evolved. Here, using phylogenetic comparative methods and data for over 1300 amphibian species, we show that egg attendance, arguably one of the simplest care behaviours, is gained and lost faster than any other care form, while complex adaptations, like brooding and viviparity, are lost at very low rates, if at all. Prolonged care from the egg to later developmental stages evolves from temporally limited care, but it is as easily lost as it is gained. Finally, biparental care is evolutionarily unstable regardless of whether the parents perform complementary or similar care duties. By considering the full spectrum of parental care adaptations, our study reveals a more complex and nuanced picture of how care evolves, is maintained, or is lost.

[1] Department of Biological and Marine Sciences, University of Hull, Cottingham Road, Hull HU6 7RX, UK. [2] Energy and Environment Institute, University of Hull, Cottingham Road, Hull HU6 7RX, UK. [3] School of Biological Sciences, Queen's University Belfast, 19 Chlorine Gardens, Belfast BT9 5DL, UK. *email: A.I.Furness@hull.ac.uk; I.Capellini@qub.ac.uk

Parental care—defined as any parental behaviour or adaptation that increases offspring fitness, often at some cost to the carer—exhibits striking diversity among species, ranging from short term and relatively simple behaviours, such as egg attendance, to long term and elaborate adaptations, like some forms of food provisioning, viviparity and lactation[1]. Not only does parental care affect the fitness of offspring and parents, but it also has profound consequences for social evolution—it leads to both cooperation and conflict within families[2–4], is associated with changes in species' life history strategies[5–7], is related to mating system and sexual selection[8] and is one of the main drivers for the evolution of sociality[9,10]. Thus, unravelling how parental care evolves and is evolutionarily maintained has important implications for our understanding of many aspects of animal life. While care behaviours and adaptations differ in complexity, duration and predicted costs and benefits for the carer and each of the sexes[1,11], we still do not know how such diversity has evolved. This is because most studies focus on one or few forms of care or reduce diversity to a coarse presence-absence species characteristic. Therefore, questions such as whether some care forms are more common because they are easier to evolve; whether short term care is a first necessary step for the evolution of prolonged care; whether the sexes differ in care form and duration, and how this in turn determines the evolutionary origin and persistence of biparental care, are still unanswered[1,11]. To answer these questions we need a holistic approach that considers many care behaviours and adaptations with regard to their function, care duration and caring sex, in a single, highly diverse taxon, since deriving generality of principles from individual case studies has proven challenging[11]. To this end, we have compiled a comprehensive dataset of parental care behaviours and adaptations in amphibians, one of the most diverse groups for reproductive and care strategies[12–15], and investigate the evolution of parental care diversity at large comparative scale.

Some forms of parental care, such as egg or offspring attendance, are much more common than others and this may be due to differences in their phenotypic complexity. Specifically, it is often suggested that some traits, including some forms of care[11], do not require major physiological, morphological or behavioural changes to evolve and thus should be gained frequently and easily in a single or few evolutionary steps. These simpler traits are also expected to facilitate the evolution of more complex traits through a progressively greater elaboration of the original trait (e.g. nuptial gifts[16]; sociality[9]; parental care[17,18]). In the context of care, egg attendance is classed as one of the simplest care behaviours because its evolution should only require that the parent(s) remain at the egg laying site after oviposition[1]. In contrast, viviparity is often considered complex since it entails numerous anatomical and physiological coadaptations between the developing offspring and the maternal reproductive tract[19], and should evolve only after internal fertilization and prolonged egg retention[20].

Likewise, the evolutionary loss of simpler traits should be easier and more frequent than that of more complex traits[21,22]. This may be particularly the case for parental care adaptations that entail profound morphological, anatomical or physiological changes in both parent(s) and offspring as this could make losing care difficult. Even simple parental care forms can affect the evolutionary trajectory of the offspring's phenotypes. For example, a predictable level of care in burying beetles (Nicrophorus vespilloides) causes the evolution of smaller mandibles in the larvae, which become less self-sufficient as they can rely on parental support for feeding[23]. If parental care leads to extensive evolutionary changes in offspring traits to the point that the offspring become highly dependent upon care for survival, losing care may no longer be a viable evolutionary trajectory because it would require crossing a major fitness valley with reduced offspring survival[21]. Thus, simpler care behaviours that offer limited opportunities for correlated evolution between parent and offspring traits, like egg attendance, should be lost more frequently and quickly than complex care forms, like viviparity. If this hypothesis is correct, it follows that simpler care forms should evolve faster and more frequently than more complex ones[1]; in other words, simpler traits should exhibit higher evolutionary rates of gain and loss than more complex traits.

Beyond the extreme cases of egg attendance and viviparity, the literature is, however, silent on how we should classify, with regard to complexity, the many other parental care forms that we find in nature. Many forms of care are likely to be more elaborate than egg attendance but not involve as many, or as profound, coadaptations between parent and offspring as in viviparity. Amphibians exhibit huge diversity in parental care behaviours and adaptations, including many that show some degree of coadaptation between parental and offspring characteristics. These range from the transport of tadpoles and juveniles, to brooding of eggs and offspring on the back or inside the parents' body (other than the oviduct), to provisioning through trophic eggs in anurans or sloughed off skin in caecilians[12–15,17,24]. Thus, rather than a simple or complex dichotomy, it is likely that these traits fall somewhere along a continuum between the two extremes of egg attendance and viviparity; if so, they should be gained and lost at intermediate evolutionary rates. To our knowledge no study to date has formally tested this prediction particularly in parental care studies and across traits of apparently different degree of complexity.

Not only the form, but also the duration of parental care is highly variable between species. Prolonged care should entail higher costs to the parents in terms of energy, time and lost mating opportunities, than temporally limited care[1]. Furthermore, once evolved, parental care may trigger selection in the offspring to increase the amount of care received, for example through behaviours such as begging[25] or hormonal manipulation of maternal resource allocation in viviparous, placental species[26–28]. Thus, early care with egg attendance in birds is considered a necessary evolutionary precursor for the evolution of longer care duration through the acquisition of chick attendance, which in turn would promote energetically expensive, parental food provisioning and lead to the evolution of prolonged care from the egg stage to nutritional independence[29]. Likewise, early care at the egg stage in amphibians is believed to facilitate the evolution of care at later developmental stages by prolonging attendance and guarding behaviour beyond hatching[17]. We should thus expect that costly, prolonged parental care should evolve in steps, starting from temporally limited care. However, whether parental care confined to the earliest stages of development leads to the evolution of prolonged parental care through the acquisition of care at the later stages has not, to our knowledge, been formally tested. Nor do we know to what extent differences in care duration between species depend on the type of care performed and on which sex cares.

Finally, a long standing question is why one parent cares for the offspring when the other does so already, i.e. why biparental care evolves. While comparative studies show that biparental care mainly arises from males joining females[30,31], within species studies reveal that biparental care is often fraught with conflict over the level of care provided by each parent, the degree of compensation for reduced partner effort, and the risk of complete desertion[32–36]. Consistent with these observations, theoretical models predict that biparental care should be evolutionarily unstable when the sexes differ even slightly in the costs and/or benefits of caring, leading to the loss of care by one parent and hence the evolution of uniparental care[34,37–40]. However,

biparental care could be evolutionarily stable if the sexes provide synergistic care through complementary division of labour[37], a still untested prediction.

Here we investigate how diversity in parental care has evolved in relation to trait complexity and function; whether shorter care (e.g. limited to the egg stage) facilitates the evolution of prolonged parental care (e.g. from the egg to the larval and juvenile stage) and if this depends on care form and caring sex, and whether biparental care is evolutionarily stable with division of labour. Amphibians are an excellent taxon in which to address these questions because, along with fishes, they exhibit the greatest diversity of parental care forms of any vertebrate class, ranging from no care to care at all stages of development (egg, tadpole, juvenile), and including uniparental male or female care, or biparental care[12–15]. Moreover, uniparental male care is present in nearly as many amphibian species as uniparental female care and, with the exception of viviparity and feeding, males may perform the same care duties as females at each stage of development[12–15,17], including complex care forms such as brooding[24]. We demonstrate that the simplest care behaviour—egg attendance—is gained far more quickly than any other care forms, while complex care adaptations—viviparity and brooding—are lost at a low rate, if at all. Furthermore, prolonged care from the egg to the juvenile stage evolves from short term care but, unexpectedly, it can just as easily revert to temporally limited care. Finally, our study reveals that biparental care, regardless of whether the parents perform the same or complementary care duties, is evolutionarily unstable, and is quickly lost to uniparental care or no care.

## Results and discussion

**Care form, complexity and function**. We have compiled a large and comprehensive dataset of parental care diversity in amphibians with information on presence or absence of care forms (attendance, transport, brooding, feeding and viviparity) at three developmental stages (egg, tadpole and juvenile) for 1322 species with no missing data (Fig. 1a; see also 'Data collection' in Methods; Supplementary Note 1, Supplementary Fig. 1, Supplementary Table 1). With this dataset we test hypotheses on the evolutionary origin and persistence of parental care using modern phylogenetic comparative approaches in a Bayesian framework ('Analysis' in Methods). We first test the hypothesis that simpler traits evolve more easily than more complex ones. Using a Reversible Jump (RJ) Multistate analysis in *BayesTraits* ('Analysis' in Methods)[41,42] this translates into the prediction that the evolutionary rates of gain and loss of simpler traits, such as attendance, are higher than those of more complex ones, such as viviparity. In support of this hypothesis, egg attendance is gained at a rate of an order of magnitude greater than all other forms of care, which evolve at similar slower rates (Fig. 1b; Supplementary Table 2a). The rate of loss is much more variable across care forms than the rate of gain. Specifically, while egg attendance is lost at a similar rate as it is gained, traits such as tadpole attendance and feeding, and juvenile attendance and transport, are lost very rapidly. On the opposite end of the spectrum, viviparity and brooding (incubating the offspring on or inside the parents' body) are lost slowly with about 16–23% of RJ Multistate models estimating their rate of loss to be zero (Fig. 1c, Supplementary Table 2b).

Parental care is more common at the egg stage than at the tadpole and juvenile stages (Fig. 1a, Supplementary Fig. 1, Supplementary Table 1a). This may partially be due to greater opportunities for care to evolve at earlier than later stages. To address this and remove the potential confounding effect of opportunity on the evolutionary rates of traits of different

complexity, we identify functionally equivalent behaviours and adaptations across stages—attendance of eggs, tadpoles and juveniles where parents remain with the offspring; transport, in which tadpoles or juveniles are moved by parents from one location to another; brooding when eggs or tadpoles develop on or within the parental body; and feeding where tadpoles or juveniles are nutritionally dependent on the mother[12–15,17]. Therefore, to further test whether phenotypic complexity explains differences in rates of evolution between traits, we repeated the analysis grouping care forms by function regardless of the stage of development at which they occur (Fig. 2a, Supplementary Table 1b for sample sizes by function). The RJ Multistate analysis confirms that attendance evolves at a much faster rate than any other care function; transport and feeding exhibit intermediate rates of gain and brooding and viviparity are gained at the slowest rate (Fig. 2b, Supplementary Table 3a). Once evolved, attendance can be lost as quickly as it is gained, while transport exhibits a moderate rate of loss; in contrast, brooding and viviparity, and to some extent feeding, are lost at low rates with 16–24% of models estimating their rate of loss as zero (Fig. 2c, Supplementary Table 3b).

While altogether these results broadly support the hypothesis that simpler parental care behaviours evolve more easily than more complex care adaptations[1], they also reveal a more nuanced picture than previously appreciated. Our finding that egg attendance can be gained quickly, strongly supports the idea that the evolutionary origin of this behaviour should only require the parent to stay at the site of egg deposition at presumably little cost[1,43], while the evolution of later elaborations that increase offspring survival, such as active defence of the clutch against predators or prevention of egg dehydration, may promote the persistence of this behaviour over evolutionary time[12–15,17]. Unexpectedly, simple care forms such as attendance at the tadpole and juvenile stages are gained at rates similar to those of complex care forms like viviparity, and at lower rates than attendance at the egg stage. Although this may indicate that complex traits may not be as difficult to evolve as anticipated, we suggest that it is more likely that some simpler care forms evolve more slowly than expected. Specifically, ecological opportunities may limit the conditions under which simpler care behaviours, such as attendance at the tadpole and juvenile stage, are selected for at later developmental stages. For example, while terrestrial clutches of eggs may be easy to defend because they are spatially localised, tadpole attendance is observed in species where the larvae remain together, such as in terrestrial nests or burrows; in larger ponds where they school together; or in small water bodies that parents keep oxygenated or connect to larger ponds by digging channels when they dry out[12–15,17].

The rate at which care forms are lost is more variable across traits than the rate of gain and supports the prediction that it is harder to lose complex traits than simpler ones. Specifically, egg attendance is lost as quickly as it is gained, suggesting that a change in selection pressures, such as higher costs of guarding, could lead to an easy loss of this behaviour. In contrast, viviparity and brooding are lost at a much slower rate, if at all, in amphibians. Brooding entails the development of offspring on or inside the parents' body and includes diverse adaptations, such as gastric brooding, brooding in the vocal sacs or brooding eggs embedded in the dorsum[12–15]. Some of these forms may rival viviparity for complexity and specialization[15,24,44,45], potentially explaining why these adaptations may not be easily lost. By investigating diversity of care in a single taxon, our study also reveals that behaviours, such as tadpole transport, are lost at relatively low rates. Albeit apparently simple, transport requires good parental spatial memory of suitable pools with low predation risk in which to release the offspring[46,47], and involves

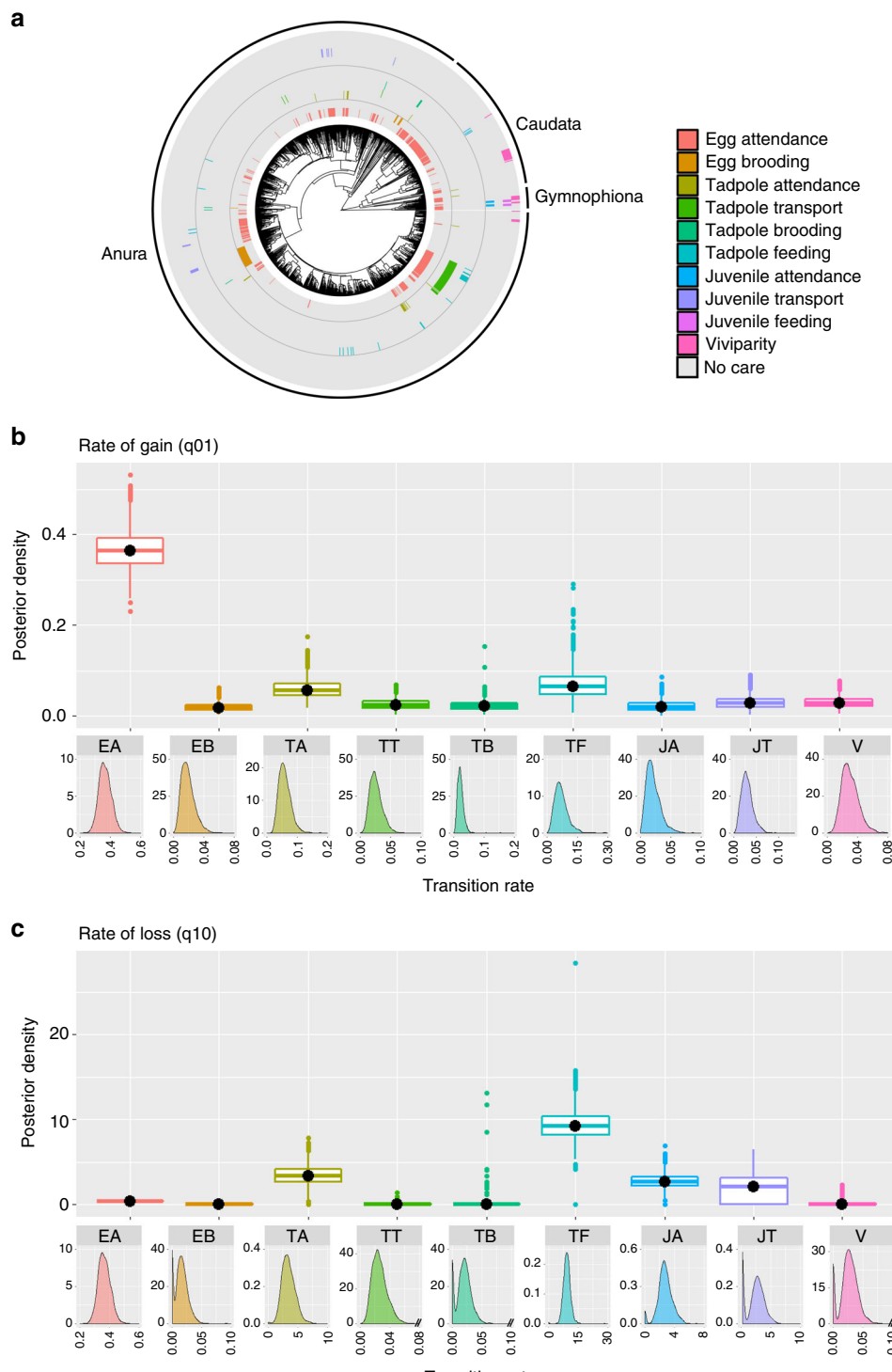

**Fig. 1** Evolutionary history of parental care diversity in amphibians. **a** Distribution of parental care forms in amphibians ($n = 1322$ species; sample sizes for each care form in Supplementary Table 1a and Supplementary Fig. 1). The posterior distributions of the RJ Multistate transitions for the **b** rates of gain ($q_{01}$) and **c** rates of loss ($q_{10}$) of each care form are shown as box plots for comparison, and as posterior density plots for each care form alone. The central black dot in the box plots indicates the median, the box the upper and lower quartiles, the vertical lines the 95% credible intervals of the posterior distributions, and the filled dots beyond the lines indicate outlier estimates. Care forms are coded with the same colours across all panels. In **b** and **c**: EA egg attendance, EB egg brooding, TA tadpole attendance, TT tadpole transport, TB tadpole brooding, TF tadpole feeding, JA juvenile attendance, JT juvenile transport, V viviparity

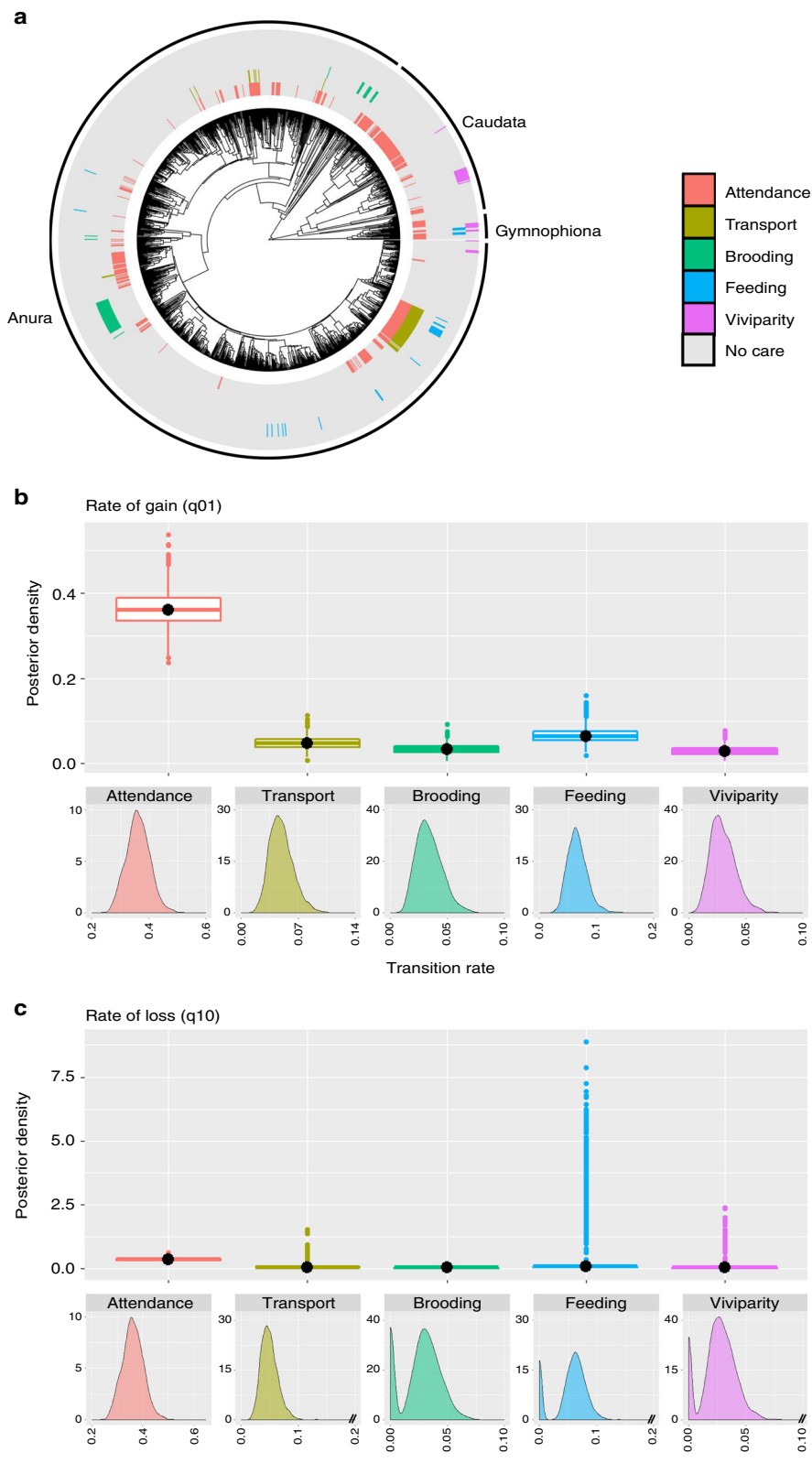

**Fig. 2** Evolutionary history of amphibian parental care grouped according to function. **a** Distribution of parental care forms in amphibians ($n = 1322$), grouped according to function (i.e. attendance, transport, brooding, feeding and viviparity) irrespective of stage of development at which they occur (sample sizes by function in Supplementary Table 1b). The posterior distributions of the RJ Multistate transitions for the **b** rates of gain ($q_{01}$) and **c** rates of loss ($q_{10}$) of each care form are shown as box plots for comparison, and as posterior density plots for each care form alone. The central black dot in the box plots indicates the median, the box the upper and lower quartiles, the vertical lines the 95% credible intervals of the posterior distributions, and the filled dots beyond the lines indicate outlier estimates. Care forms are coded with the same colours across all panels

some level of parent-offspring conflict and sibling competition when parents can only take one or few offspring at a time[47–49]. Together, these characteristics may explain why transport exhibits moderate rates of evolution.

**Care stage and duration.** Next, we test with RJ Discrete analysis in *BayesTraits* ('Analysis' in Methods)[50,51] whether parental care at the egg stage facilitates the acquisition of care at the larval and juvenile stages, thus promoting the evolution of prolonged care and higher parental investment. We also investigate to what extent this is influenced by care form and caring sex. We first group care behaviours within stage of development as early (egg stage) or late (tadpole and juvenile stages combined; Fig. 3a). Care adaptations covering both stages (e.g. viviparity, egg attendance and egg brooding with direct development) are classed as both early and late care ('Data collection' in Methods; Supplementary

Fig. 1). We find strong support for correlated evolution between early and late care (RJ Dependent vs RJ Independent model: Log BF = 186.5), as predicted. If the hypothesis that early care is a necessary precursor for the evolution of prolonged care is correct, we also expect to identify a two-step evolutionary pathway from no care to early care first, and from early care to prolonged care through the acquisition of care at later developmental stages (Supplementary Fig. 2a). Furthermore, if prolonged care is an evolutionary stable strategy that is harder to lose once evolved, the transition rates leading to the evolution of prolonged care from short term care should be higher than those in the opposite direction (Supplementary Fig. 2a). We thus examine the magnitude of the transition rates estimated by the RJ Discrete Dependent model to evaluate how results conform to these predictions.

The RJ Dependent models shows that parental care at the egg stage is equally likely to evolve as it is to revert back to no care,

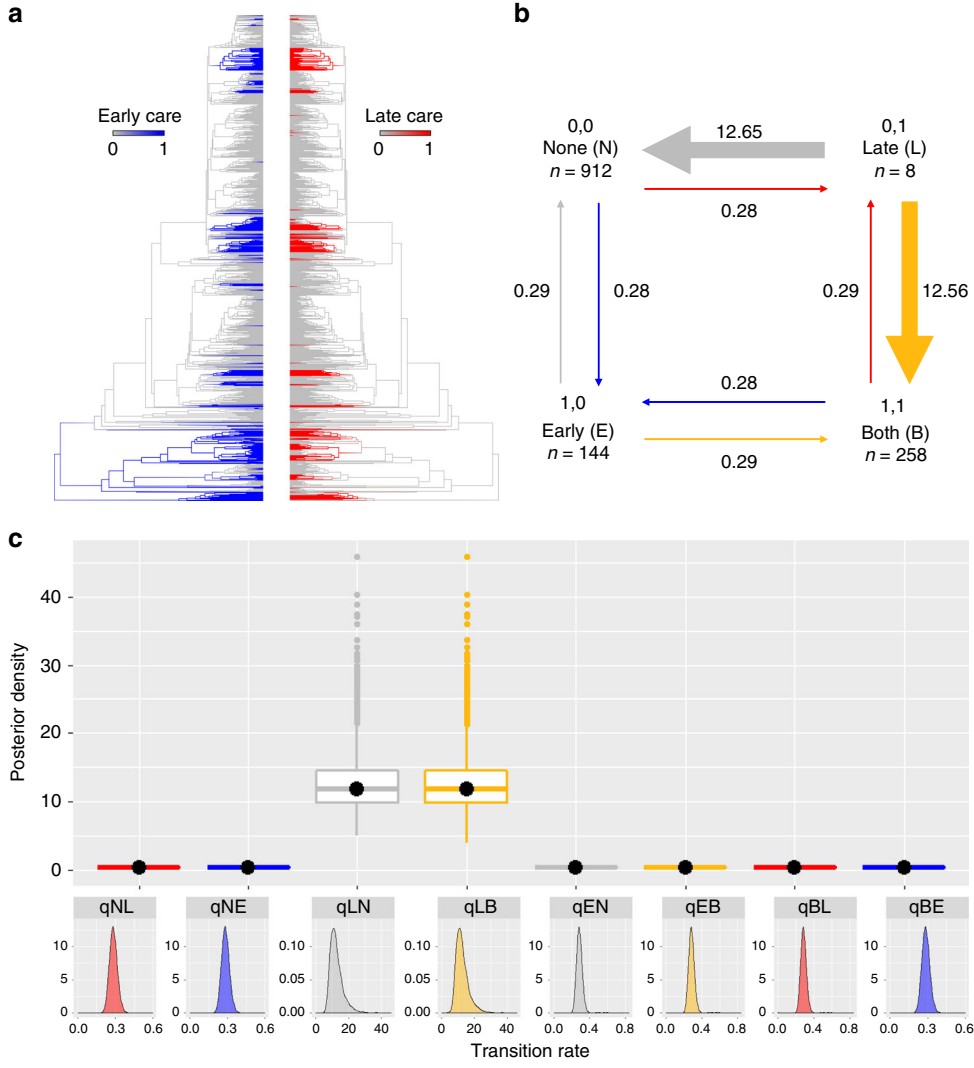

**Fig. 3** The correlated evolution of early and late care. **a** Early care (blue) and late care (red) plotted on the phylogeny using stochastic character mapping with the R package *phytools*[73]. **b** Summary diagram of the transition rates across the four combinations of character states from the RJ Discrete Dependent model of evolution. The sample sizes for each combination of character states are reported; the arrows are scaled to reflect the magnitude of mean transition rates from the posterior distribution, with the mean value also indicated. **c** Posterior distributions of the transition rates from the RJ Discrete Dependent model of evolution are shown as box plots for comparison and as posterior density plots for each transition rate alone. The central black dot in the box plots indicates the median, the box the upper and lower quartiles, the vertical lines the 95% credible intervals of the posterior distributions, and the filled dots beyond the lines indicate outlier estimates. Transition rates among character states in **b** and **c** are indicated as follows: no care to late only care (NL, red); no care to early only care (NE; blue); late only care to no care (LN; grey); late care to prolonged care, i.e. both late and early (LB; yellow); early only care to no care (EN; grey); early only care to prolonged care, i.e. both late and early (EB; yellow); prolonged care to late only care (BL; red); prolonged care to early care (BE; blue)

and it can as easily facilitate the evolution of prolonged care through the acquisition of care at later developmental stages (Fig. 3b, c, Supplementary Table 4a). We also find qualitatively similar results when we repeat the analysis by parental sex (Supplementary Table 4b, c), indicating not only that both sexes are equally likely to evolve early care from no care but also that they both can increase parental investment by caring at later stages of offspring development. Thus, in both sexes early care is an evolutionarily dynamic state that can be gained easily from the lack of care, can be lost as easily as it is gained or can as easily lead to an increase in care duration and the evolution of prolonged care.

When we consider individual care forms at different developmental stages with sufficiently large sample sizes for analysis, we find care at the earlier developmental stage promotes care at the tadpole or juvenile stages, with equally likely reversals to short term care and no care. Specifically, egg attendance without direct development is evolutionarily associated with and promotes the evolution of tadpole attendance (RJ Dependent vs RJ Independent model: Log BF = 53.8; Supplementary Table 5a) and tadpole transport (RJ Dependent vs RJ Independent model: Log BF = 23.0; Supplementary Table 5b), but not tadpole feeding (RJ Dependent vs RJ Independent model: Log BF = −0.1). Tadpole feeding is typically present when the larvae develop in water-filled, nutrient poor cavities within terrestrial plants. The limited availability of food resources in such cavities has probably led to the evolution of females repeatedly visiting them to deposit eggs for their larvae to consume; in some cases maternal provisioning is essential for tadpole survival and successful metamorphosis[48,52–54] and tadpoles exhibit specialized morphology for oophagy[55]. Presumably these cavities afford some protection against predation and desiccation[48,52–54], which could make egg attendance superfluous and explain why tadpole feeding may evolve in the absence of egg attendance. Next, we investigated which care forms promote the evolution of care at the juvenile stage. We find that juvenile care is only associated with and promoted by the evolution of egg attendance with direct development (RJ Dependent vs RJ Independent model: Log BF = 28.1; Supplementary Table 5c), and not by egg brooding with direct development or by viviparity (egg brooding with direct development and juvenile care: Log BF = 1.3; viviparity and juvenile care: RJ Dependent vs RJ Independent model: Log BF = −4.9).

Altogether, our results suggest that temporally limited, early investment in parental care can lead to higher, prolonged care in both sexes through the acquisition of care behaviours at the tadpole and juvenile stages, as predicted, and egg attendance is a major driver of care at later developmental stages. We suggest that, once early care has evolved, parent-offspring conflict and sibling competition may be particularly effective in promoting a further increase in parental investment and care duration through mechanisms such as begging and hormones[21,25,27]. For example, some anuran tadpoles perform solicitation displays analogous to begging that induce parental feeding and transport[48,56]. These displays may offer tadpoles opportunities to influence parental behaviour and should be favoured in species with clutches larger than a female can feed or parents can transport, as this creates intense sibling competition and parent-offspring conflict[47–49,57]. Furthermore, viviparity and brooding—which are lost slowly, if at all, in amphibians (Fig. 2b)—may allow the offspring to exert some control over parental resource allocation through release of hormones, as they do in placental mammals[26–28,58]. While we still lack in depth knowledge of the physiology of these adaptations, exchange of nutrients and chemicals between parents and offspring has been reported in brooding anurans[24,44,45]. Given the permeable nature of amphibian skin, offspring

manipulation of parental care levels may be facilitated in species that brood the eggs on the back, covered by parental skin, as this could allow the developing offspring easy access to parental blood vessels.

Surprisingly, our analysis also reveals that a reduction in the duration of care from prolonged to early care is not only possible but it is as likely as the acquisition of late care (Fig. 3b, c), irrespective of parental sex (Supplementary Table 4b, c). Once egg attendance has evolved, tadpole transport and tadpole attendance can be as easily gained as lost; likewise care at the juvenile stage can be quickly lost in direct developing species (Supplementary Table 5). Most theoretical models on the duration of parental care focus on biparental species and how sexual conflict and mating opportunities determine when a parent should desert the offspring[3]. While these models have advanced our understanding of family dynamics in biparental species, we still lack theoretical predictions on how prolonged parental investment could be reduced over evolutionary time in uniparental species. We propose that a reduction of care duration depends on the type of care provided and is more likely with behaviours or adaptations that enable parents, rather than offspring, to have a greater control over care levels and resource allocation. Future models could investigate how diverse forms of care, such as attendance, transport, brooding or viviparity, that offer offspring very different opportunities to influence parental resource allocation, influence the evolutionary trajectory of parental care duration, both towards higher and prolonged as well as lower and shorter care duration.

**Caring sex and division of labour.** Finally, we address the long standing question of how biparental care has evolved and investigate how this is affected by care form and division of labour. If biparental care is an evolutionary unstable condition as predicted by theoretical models[34,37–40], we expect that it is lost faster than it is gained; hence, under our analytical framework, transition rates 'out' of biparental care towards uniparental care by either sex should be higher than those leading to the evolution of biparental care (Supplementary Fig. 2b). Our RJ Discrete analysis in *BayesTraits* ('Analysis' in Methods)[50,51] finds strong support for the Dependent model of evolution (RJ Dependent vs RJ Independent Discrete model: Log BF = 5.9; Fig. 4a), suggesting that the evolution of parental care in one sex (gain and loss) influences the evolutionary trajectory of care in the other sex. The magnitude of transition rates of the RJ Dependent model reveals that male and female uniparental care are gained from and lost back to no care at approximately equal, low rates (Fig. 4b, c, Supplementary Table 6a). In contrast to insects and other terrestrial vertebrate classes where biparental care is more likely to arise when males join caring females[30,31], biparental care in amphibians evolves at approximately equal rate from either uniparental male or female care. Once evolved, biparental care shows high rates of transition back towards male only or female only care, indicating that it is an evolutionarily unstable condition (Fig. 4b, c, Supplementary Table 6a). We find similar results when we study the evolutionary stability of biparental care for individual behaviours for which we have sufficiently large sample sizes. Specifically, uniparental egg attendance evolves at low rate in both sexes and towards biparental egg attendance, but the latter reverts quickly to uniparental care (Supplementary Table 6b). Biparental tadpole transport arises primarily from male only transport but it is as quickly lost to uniparental female transport and, to a lesser degree, to uniparental male transport (Supplementary Table 6c). For attendance and transport both sexes are probably equally effective carers and the benefits to the offspring of having two caring parents are likely non-additive[12,13]. This may explain not

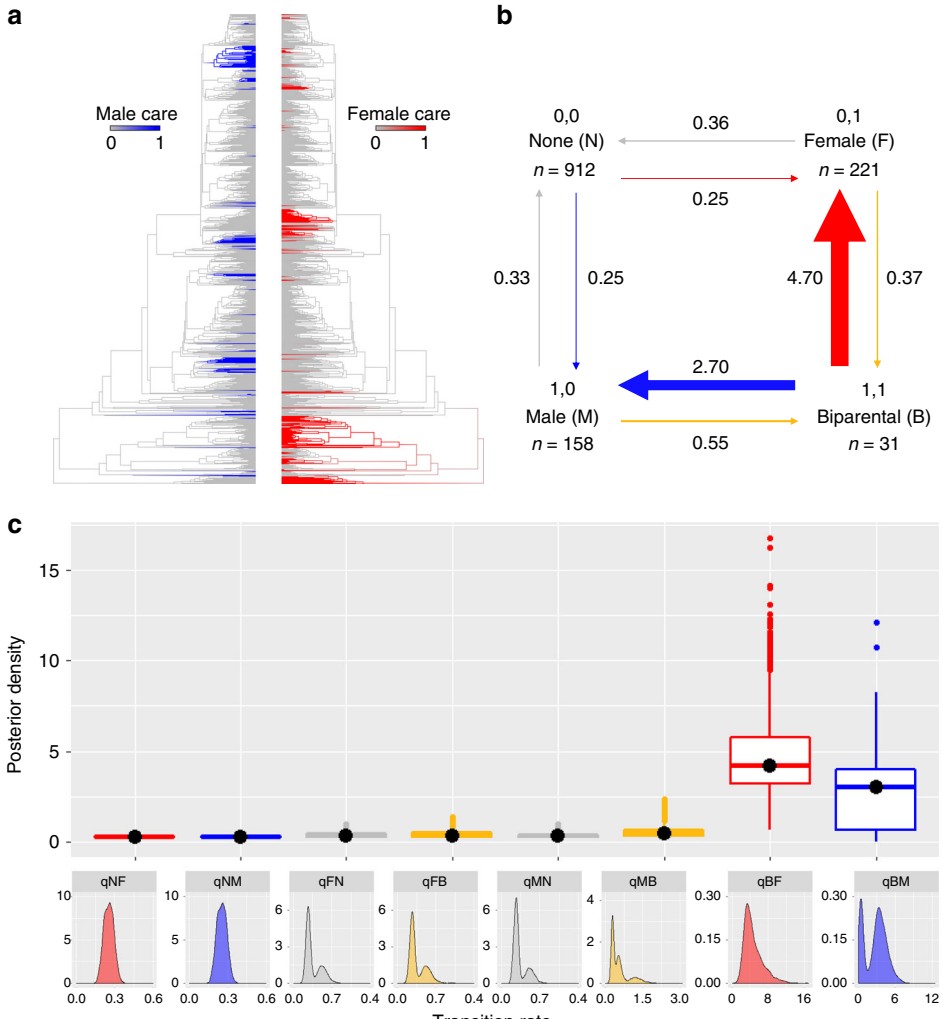

**Fig. 4** The correlated evolution of male and female care. **a** Male care (blue) and female care (red) plotted on the phylogeny using stochastic character mapping with the R package *phytools*[73]. **b** Summary diagram of the transition rates across the four combinations of character states from the RJ Discrete Dependent model of evolution. The sample sizes for each combination of character state are reported; the arrows are scaled to reflect the magnitude of mean transition rates with the mean value also indicated. **c** Posterior distributions of the transition rates from the RJ Discrete Dependent model of evolution are shown as box plots for comparison and as posterior density plots for each transition rate alone. The central black dot in the box plots indicates the median, the box the upper and lower quartiles, the vertical lines the 95% credible intervals of the posterior distributions, and the filled dots beyond the lines indicate outlier estimates. Transition rates among character states in **b** and **c** are indicated as follows: no care to female only care (NF, red); no care to male only care (NM; blue); female only care to no care (FN; grey); female only care to biparental care (FB; yellow); male only care to no care (MN, grey); male only care to biparental care (MB; yellow); biparental care to female only care (BF; red); biparental care to male only care (BM; blue)

only the evolutionary instability of biparental care but also why, in the few species in which both sexes partake in attendance or transport, they rarely perform these duties simultaneously or equally[12,14]. Instead, one sex or the other seem to be the sole or primary caregiver within a pair, with differences among pairs in whether it is the male or the female[59,60]. Overall, these results support theoretical models suggesting that biparental care is an evolutionarily unstable state due to different costs of care between the sexes and selection for desertion[34,37–40], for example in search of new mating opportunities, locking the other sex into uniparental care[37].

However, biparental care is predicted to be evolutionarily stable and maintained when parents provide complementary care, i.e. with division of labour[34,37,39,40]. To test this hypothesis, we classify species as either exhibiting no biparental care (*n* = 1291), overlapping biparental care (when the sexes undertake the same care behaviour at the same stage of development; *n* = 14) or complementary biparental care (when males and females perform

distinct behaviours or at least some behaviours are performed solely by one sex at a given stage of development; *n* = 17). If division of labour makes biparental care an evolutionary stable condition, we expect that, under our analytical framework, complementary care is lost at a lower rate than it is gained, and the opposite for overlapping care (Supplementary Fig. 2c). RJ Multistate analysis in *BayesTraits* ('Analysis' in Methods)[41,42] reveals that overlapping and complementary biparental care evolve at approximately equal, low rate from the absence of biparental care, but both forms of biparental care are quickly lost to no biparental care, suggesting that neither is evolutionarily stable (Fig. 5, Supplementary Table 7). Transitions between the two types of biparental care are also unlikely to happen, given that 41–51% of RJ models estimate their rates of evolution as zero (Fig. 5, Supplementary Table 7). Altogether, our results strongly indicate that biparental care, in whatever form it arises, is an evolutionarily unstable condition that is quickly lost. Besides a division of labour, other factors are believed to promote the stability of biparental care. These include

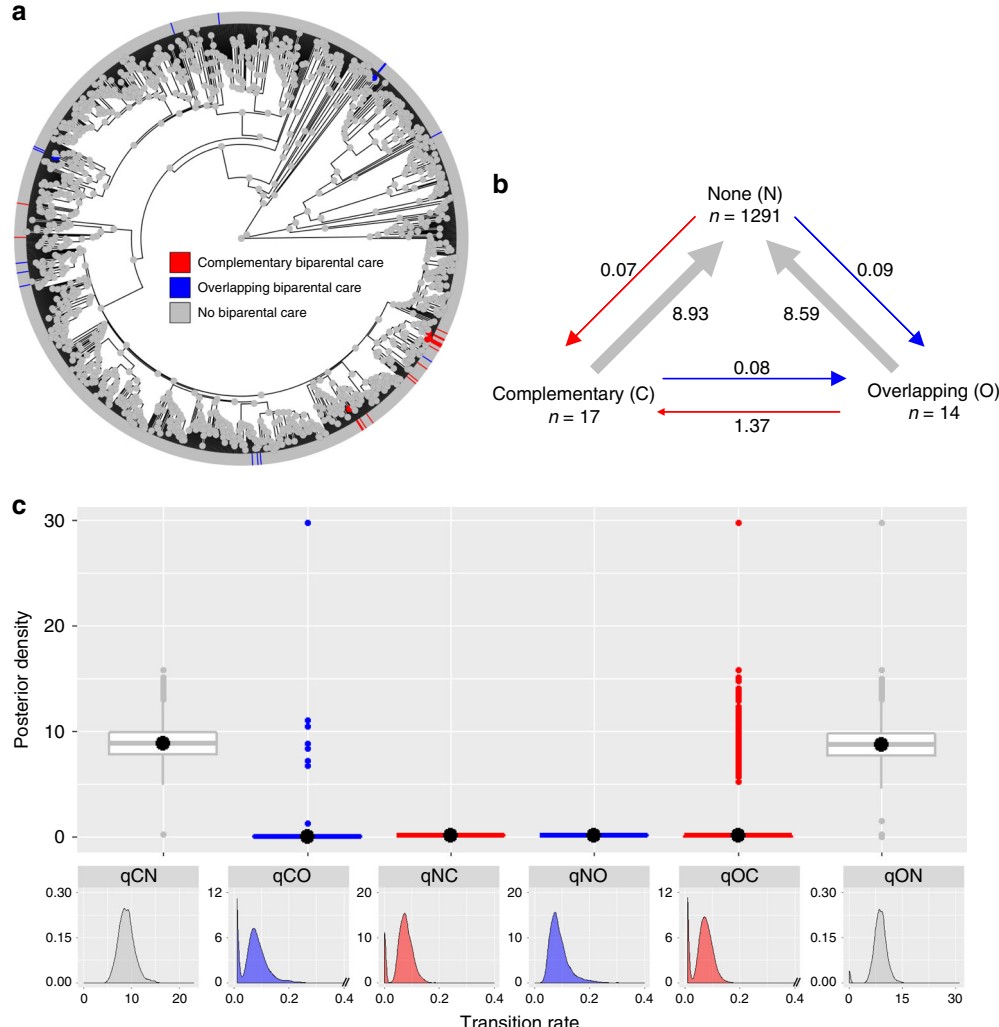

**Fig. 5 The evolution of biparental care forms. a** Overlapping and complementary care plotted on the phylogeny using stochastic character mapping with the R package *phytools*[73]. **b** Summary diagram of the transition rates across the three possible character states of biparental care from the RJ Multistate model of evolution. The sample sizes for each character state are reported; the arrows are scaled to reflect the magnitude of mean transition rates with the mean value also indicated. **c** Posterior distributions of the transition rates from the RJ Multistate model of evolution are shown as box plots for comparison and as posterior density plots for each transition rate alone. The central black dot in the box plots indicates the median, the box the upper and lower quartiles, the vertical lines the 95% credible intervals of the posterior distributions, and the filled dots beyond the lines indicate outlier estimates. Transition rates among character states in **b** and **c** are indicated as follows: complementary to no biparental care (qCN, grey), complementary to overlapping biparental care (qCO, blue), no biparental care to complementary care (qNC, red), no biparental care to overlapping care (qNO, blue), overlapping to complementary biparental care (qOC, red) and overlapping to no biparental care (qON, grey)

environmental conditions in which offspring survival is substantially reduced with a single caring parent, or in which future reproductive opportunities are limited so that desertion is unlikely to increase the parents' fitness compared to caring[22,34,38,61]. Future studies could test these hypotheses in amphibians once data for a large number of species become available.

## Conclusions
To sum up, by incorporating the full spectrum of parental care adaptations, our study reveals that different forms of care are gained and lost at different rates. We also demonstrate that prolonged care can be gained from and lost back to temporally limited care in both sexes, and that biparental care is an evolutionarily unstable state in whatever forms it arises. Altogether our results suggest that reduction in care duration—from prolonged to short term or no parental care—depends on the nature of care forms. Specifically, we propose that reduction in care levels, or the

complete loss of care, is more likely when parents have greater control over resource allocation, such as in attendance. Conversely, reducing care and parental investment is more difficult once adaptations that trigger evolutionary arms races between parent and offspring or between siblings evolve, like brooding and viviparity, as these offer greater potential for the offspring to manipulate resource provisioning[27,28]. Such arms races may ultimately lead to tightly correlated changes in parent and offspring phenotypes so that the loss of these adaptations, and consequently parental care, becomes unlikely. We anticipate that similar results to those presented here may be found in other taxonomic groups that show comparable diversity in parental care to the amphibians, such as insects[22,30] and fishes[20,34,40]. More broadly, considering the whole diversity of behaviours—such as diverse forms of sociality, reproductive mode or courtship—may unravel unexpected evolutionary pathways and provide fundamental insight on the selection pressures that have led to their origin and maintenance.

## Methods

**Data collection.** We classified parental care separately by developmental stage (Supplementary Fig. 1). Care forms included attendance at the egg, tadpole and juvenile stages; transport at the tadpole and juvenile stages; brooding at the egg and tadpole stages; feeding at the tadpole and juvenile stages and viviparity (Supplementary Note 1 for detailed definitions). We recorded the presence or absence of each parental care behaviour, and if present, the caregiver sex (Supplementary Data 1). We extracted and cross-checked information on parental care from reviews, supplemented with data from 323 primary and secondary references (Supplementary Data 2). The absence of a behaviour was rarely mentioned in primary sources, making it difficult to distinguish between species that genuinely lack parental care from those in which it was not reported due to limited research effort. To address this, we developed a multifaceted approach to scoring no care. First, we scored parental care as absent for a given species if it was scored as such in one or more of six recent large comparative datasets of amphibian parental care[55,62–66]; where present, we resolved any discrepancies among these six sources by consulting the primary literature. Next, we searched Web of Science and Google Scholar for all amphibian species in the phylogenetic tree[67] lacking any information on parental care (either presence or absence) and checked any articles with parental care, reproductive mode, life history, mating, natural history or reproduction in the title (Supplementary Note 1). We classified species as lacking parental care if we found clear statements of no care, or if there was no mention of any care behaviours or care adaptations in sources that otherwise described in detail the reproductive biology of the species. All remaining species in the phylogeny for which we either could not find any evidence of parental care or could not confirm the absence of care following the procedure here described, were discarded ($n = 1987$). Our final dataset included 1322 species with no missing data across all care forms; of these 410 exhibited parental care and 912 lacked any form of care (Supplementary Table 1; Supplementary Data 1).

From these data we classified parental care by function irrespective of stage, and by stage (i.e. early or late) irrespective of function. Functional groupings included attendance, transport, brooding, feeding and viviparity, regardless of the stage of development (i.e. egg, tadpole or juvenile) at which they occurred. With regard to stage, we defined early care as care at the egg stage, thus species with egg attendance, egg brooding, or viviparity were scored as exhibiting early care (Supplementary Fig. 1). We defined late care as care at the larval or juvenile stage (Supplementary Fig. 1). Egg attendance and egg brooding included species with direct development, in which the offspring hatch as juveniles, and species with a larval stage (in which the offspring hatch as tadpoles that become free living; Supplementary Fig. 1). Thus, late care included egg attendance and egg brooding in direct developing species, tadpole attendance, tadpole transport, tadpole brooding, tadpole feeding, juvenile attendance, juvenile transport, juvenile feeding and viviparity (Supplementary Fig. 1). Note that, therefore, care forms that cover more than one stage (e.g. viviparity, egg attendance and egg brooding with direct development) were classed as both early and late care. For the analysis of care duration considering different care forms across stages, we divided egg attendance and brooding into species with and without direct development and tested for the correlated evolution between egg attendance without direct development and tadpole attendance, tadpole transport and tadpole feeding; egg attendance with direct development and juvenile care; egg brooding with direct development and juvenile care; and viviparity and juvenile care. In analyses at the juvenile stage, juvenile care forms were classed together as sample sizes of separate care forms at this stage were too small (Supplementary Fig. 1, Supplementary Table 1). Likewise, the sample size was too small for analysis between tadpole brooding ($n = 7$ species) and care forms at the earlier or later stages. No species in our dataset exhibits egg brooding without direct development and care at the tadpole stage, or tadpole care and juvenile care; therefore, associations between these traits were not tested.

Finally, for the analysis of biparental care we classified biparental species as exhibiting overlapping care if the sexes perform the same care behaviour at the same stage of development ($n = 14$). Conversely, we considered complementary care if the sexes care at different stages or where at least some behaviours are performed solely by one sex at a given stage of development ($n = 17$). For example, in some *Nyctibatrachus*[68] frogs, both parents attend the developing eggs at the oviposition site[68] (i.e. overlapping biparental care). In contrast, some Dendrobatid frogs exhibit complementary biparental care; for example, the male attends the eggs, while the female transports the tadpoles to water-filled plant cavities and later deposits trophic eggs for the tadpoles to consume[69,70].

**Analysis.** To investigate the evolution of parental care we used Multistate and Discrete models in *BayesTraits* V3[42,51] in a Bayesian framework with a comprehensive, dated, molecular phylogeny of Amphibians[67]. We scaled the tree, that did not have any polytomies, by a constant using the default setting of a mean branch length of 0.1 as recommended in the *BayesTraits* manual. Note that scaling the branch lengths of the tree does not alter results and conclusions as the procedure also scales the parameter space of the transition rates by the same constant; scaling the tree, however, allows the algorithm to better explore parameter space when rates are very small and hard to estimate or to search for. All analyses employed Reversible Jump (RJ) Markov Chain Monte Carlo (MCMC) with an exponential prior whose mean was seeded from a uniform hyperprior ranging from 0 to 20, or 0 to 100 (details for each analysis in Supplementary Tables 2–7). Models are

sampled in direct proportion to their fit to the data in MCMC and the RJ procedure enables a reduction of model complexity and over-parametrization by setting some transition rates equal to zero or equal to one another[41]. Therefore, RJ is particularly suited to accommodate analyses with relatively small sample sizes. For a fuller and correct interpretation of results of RJ models, therefore, we need to consider not only the mean of the posterior distributions as these may not be normally distributed, but also the mode of the posterior distribution and the percentage of models in which a given parameter is estimated to be equal to 0. MCMC chains were run for 400 million or 1 billion iterations with a burnin of 500,000 and sampling every 200,000 iterations (details for each analysis in Supplementary Tables 2–7). Visual inspections of traces of all parameter estimates in *Tracer* v1.6[71] confirmed that all chains had adequate mixing and reached convergence, with all parameters having effective sample sizes greater than 1000. All MCMC analyses were run in triplicate and independent runs always produced qualitatively similar results; here we present results for the first chain.

With MCMC RJ Multistate[41,42] we investigated the evolutionary history of individual care behaviours and care by function. Multistate estimates transition rates among multiple alternative character states of a single discrete trait (e.g. the rate of transition between no care and parental care) over the phylogeny. We compared the posterior distributions of the rates of gains and losses among different behaviours or care function to investigate whether simpler forms of care are gained and lost more quickly than more complex ones. Likewise, we compared the posterior distributions of the rates of gain and losses of different types of biparental care (i.e. no care, complementary or overlapping care) to test the hypothesis that biparental care is evolutionarily unstable unless the sexes perform complementary—as opposed to overlapping—behaviours[37]. We thus expect that complementary care is lost at a lower rate than it is gained if it is an evolutionarily stable state, while we expect the opposite for overlapping care if it is an unstable state (Supplementary Fig. 2c).

We used MCMC RJ Discrete Independent and Dependent models[50,51] to evaluate the evolutionary pathways by which biparental care originates and is maintained, and to test the hypothesis that short term care leads to the evolution of prolonged care. Discrete models require two binary traits (e.g. presence/absence of female care; presence/absence of male care). Under the Independent model the two traits evolve independently of one another and the model estimates four transition rates (the rate of gain and loss for each trait). Conversely, the Dependent model estimates eight transition rates between the four combinations of character states (presence/absence) that the two binary traits can jointly take (i.e. both absent, male but no female care, female but no male care, biparental)[50,51]. If the Dependent model fits the data better than the Independent model, the two traits (e.g. male care, female care) evolve in a correlated fashion. We compared the fit to the data of Independent and Dependent models by estimating their marginal likelihood using a stepping stone sampler in MCMC in *BayesTraits*, and set the sampler to have 200 stones and 200,000 iterations per stone. The marginal likelihood of a model is its likelihood scaled by the prior probabilities and integrated across all parameter values of the posterior distribution. We then calculated Bayes Factors (BF)[41] as twice the difference in the logarithm of the marginal likelihood scores of the two competing models; BF can thus be considered the Bayesian analogue of the likelihood ratio test in a maximum likelihood framework[51,72]. BF greater than 2 are considered positive evidence that the model with the higher marginal likelihood fits the data better than the alternative model with lower marginal likelihood, greater than 5 as strong evidence, while values over 10 as very strong evidence[51]. When supported as the best fitting model, the Dependent model can also reveal whether there are preferential evolutionary pathways from the absence to the presence of both traits (e.g. female care or male care evolving first), and whether a given combination of character states for the two variables (e.g. biparental care) is evolutionarily stable (i.e. when the rates of gains are higher than the rates of reversals).

**Reporting summary.** Further information on research design is available in the Nature Research Reporting Summary linked to this article.

## Data availability
The dataset compiled and analysed for this study is included in this published article and its supplementary information files (Supplementary Data 1 and 2).

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

## Acknowledgements

We thank the University of Hull for funding to IC to support this research; James Gilbert, Lesley Morrell, Stephanie McLean and Yannis Dimopoulos for suggestions on this study, and Chris Venditti, Katharina Wollenberg Valero, Chiara Benvenuto, Ylenia Chiari, Mark Pagel and Andrew Meade for comments on early results and drafts of this manuscript. We also thank VIPER High Performance Computing facility and its support team at the University of Hull.

## Author contributions

Both authors designed the study; A.I.F. collected and analysed the data under the guidance of I.C.; both authors wrote the manuscript.

## Competing interests

The authors declare no competing interests.
