## [Peer Review File · Nature Communications]

Reviewers' Comments:

Reviewer #1:

Remarks to the Author:

Overall, I found this to be an important and interesting study well-worth publishing. While other studies of the evolution of parental care and reproductive strategies in amphibians have been done, this is the most extensive one that I know of. The authors have accumulated an impressive dataset that will, I think, be useful to comparative evolutionary biologists for years to come. Beyond this, the authors have been able to focus on what (in my opinion) are interesting questions that have not been adequately addressed in a comparative framework. The idea that simple characters should evolve more rapidly and be more volatile than complex characters is a concept of general interest in evolutionary biology, and the authors are able to address this issue in a convincing manner with their dataset. The stability of biparental care is another issue of general interest to both theoreticians and empiricists, and the authors of this study are able to provide some novel insights in this regard. The methods used are current, and appear appropriate to the questions they were used to address. I had just a few minor comments that should be addressed before publication, as noted below.

Line 67: The claim that behaviors that cannot easily be classified as simple or complex should evolve at intermediate rates seems somewhat vague. I think it might be worth specifying exactly which behaviors those are, and making a specific case for each one.

Line 187: I would be curious to know if the results remain (qualitatively) the same if behaviors (like viviparity) that span early to late stages were classified as later (I think a case could be made that this would be a logical category for that type of behavior).

Line 370: Change "allows to reduce model complexity and over-parametrization" to "enables a reduction of model complexity and over-parametrization"

Reviewer #2:

Remarks to the Author:

In this paper Furness and Capellini report on evidence of parental care across over 1000 species of amphibians about 40% of which have been recorded as displaying some form of parental care. They use this data set to carry out several phylogenetic comparative analyses to elucidate the origins and evolutionary diversification of parental care. Specifically, they focus on three main hypotheses, a) that simple forms of parental care are more evolutionarily labile than complex forms of parental care, b) that care during the early stages of juvenile development is a precursor for care at later stages of juvenile development and c) that biparental care will exhibit greater evolutionary stability when parents vary in what form of care they exhibit. They find some support for the first two hypotheses but limited support for the last hypothesis.

Overall this is an impressive data set that has the potential to contribute to our growing understanding of the origins and evolution of parental care. Amphibians are particularly tractable systems for exploring these questions because of the considerable diversity of parental care exhibited (functionally, taxonomically and in terms of which sex provides the care) providing opportunities to develop and test clear hypotheses about the factors responsible for that variation. The paper goes some way to addressing some of these questions and will no doubt provide a basis from which future hypothesis driven tests of the mechanisms underpinning some of the patterns elucidated could be undertaken. However, I had several questions regarding the conceptual basis of the paper that I think need to be addressed before the paper can really deliver on its promise. Furthermore, there are

several key aspects of the analysis that need to be addressed before the results can be interpreted with confidence. Below I detail these concerns.

Major Comments

My major comment regards the overall conceptual framework from the paper with respect to the specific aim of understanding patterns underlying the origins and evolution of parental care. This is a substantial challenge and one which the amphibian data set is potentially well established to test – however, I am not sure the framework of your paper is really set up appropriately to explore this question in the most powerful way possible. The paper at the moment is set up to explore what read at time as three disparate aims that are of relevance, broadly, to understanding the evolution of parental care. These aims are all interesting and important to address, but I think they probably want to be brought together so that the paper has a single narrative to it. At the moment it appears, in places, that the paper is there to tick off a number of understudied areas in parental care research rather than to use the data to understand evolutionary patterns of parental care in amphibians in a holistic manner. To address this I think that the paper could perhaps be split up into three sections – the first is understanding the origins and elaboration of care across different life history stages, the second is understanding the elaboration of care across functional context within life history stages, and the third is a question about the extent to which these differ between males and females and the evolution of bi-parental care more broadly. I think this would allow you to provide a more powerful and structured analysis of the evolutionary patterns of parental care diversification than you currently report. Indeed, at present the initial major focus is on the evolutionary lability of simple vs complex traits. This is of course relevant to the data on parental care that has been collected for this paper but is perhaps represents a more general question than the one being posited here. Instead I think that this aspect of the paper would be better incorporated into a more explicit focus on parental care itself where the results in terms of variation in the evolutionary lability of traits emerges as an outcome of the models rather than as an explicit focus.

To address these concerns, I would suggest a slight change in overall focus. I think the major focus should be on the extent to which simple forms of early parental care (care at the egg stage, viviparity) sets the stage for prolonged parental care at the larval and juvenile stage. This is the key question that your data allows you to address – given that you do not have predictors for what drives the initial emergence of care itself. When doing this, I would also suggest simplifying this analysis somewhat. That is, I would first simply ask whether care at the larval/juvenile stage is more likely to evolve from a background of egg care/viviparity or no care. To achieve this, I wonder if you need the diversity in care types for this initial step and instead just include the presence and absence of care at each of those stages. Instead you could focus on a simple framework like this (apologies if this is slightly oversimplified):

- a) Egg Care --- larval Care --- Juvenile Care
- b) Egg Care (with direct development --- Juvenile Care
- c) Viviparity --- Juvenile Care

You would then be interested in asking very simple questions regarding the extent to which Juvenile care evolves as a function of any earlier attendance (e.g., combine a, b and c) or whether this differs for the different early stage care backgrounds (separate analyses for a, b and c). This would then allow you to not only ask whether parental care at the juvenile stage evolves as a function of parental care at earlier stages but also whether the context of that early care matters (similar to the recent analysis done for reptiles). You could also do a smaller separate analysis for the extent to which egg care predicts larval care.

Combined I think this would provide a more robust analysis of how parental care evolved and the extent to which care in the later stages is dependent on care in the early stages (and how that differs depending on the nature of that early care). Interestingly, as your data suggests, it would also allow you to compare this to instances where the evolution of care in early vs late stages represent independent evolutionary events. This would provide a nice framework for a discussion (and potentially future analyses) on the factors that might have promoted this (e.g., there is different selection on care at the different stages). There are some analogous systems in which these questions have been studied. For example, Michael Griesser's work on the evolution of family living and cooperation in birds (PlosBiol 2017).

Importantly, the above approach would also then allow you to explore the question of the extent to which parental care elaborates within each of these care types. That is, do more complex forms of parental care evolve from simple forms of care. So, within these care types, does egg attendance provide a precursor for the evolution of brooding, feeding, transport or do those traits emerge independently (although see requirement for clarification on this point below). In this context questions about the lability of simple vs complex traits perhaps becomes more relevant as it allows you to say something about the evolutionary lability of simple attendance across these different stages and then how this has provided the basis from the evolution of further elaboration of care - and how that subsequently influences the maintenance of care.

Re-structuring the analysis in this way would also help clarify the role that traits such as viviparity play in this process. The complication is that viviparity can be viewed as both an evolved parental care strategy as well as a fundamental precursor to greater elaboration of care types. You seem to focus largely on the former in this instance (e.g., you state that viviparity is an extended form of parental care to occurs across life stages) but the above would allow you to use viviparity more powerfully to explore some of the precursors to extended forms of parental care. Indeed, one of the benefits of amphibians is that they exhibit multiple parity forms that provide fundamentally different associations between mothers and their offspring - which may have implications for the subsequent evolution of prolonged care (as well as other traits - as articulated in the introduction).

Minor Points

Introduction

Page 3, Lines 42: Minor point but I would finish the first paragraph here and start a second paragraph from "While care behaviours...". I think this would provide a more powerful introduction to the paper.

Page 3, Lines 43 - 46: This is a long sentence I would consider splitting it in two.

Page 3, Lines 43 - 46: I would also argue that the main reason for this is that in few systems is there the requisite diversity of care types to allow for a proper analysis of these traits.

Page 5, Lines 80 - 83: Is there data on these traits in offspring that could be tested in the context of the predictors of the loss of different care types? This may be outside the scope of the study, but it seems to me that this data set may also allow you to explore the extent to which the evolution of different care types have resulted in the evolution of offspring traits (perhaps focussing on the evolution of increasingly altricial offspring) - which may help explain the lability of different care types.

Page 6, Lines 105 - 106: Does this not depend on the sex that has predominantly evolved to care for the offspring? For example, in systems where there is predominantly female only care bi-parental care

occurs when males also stay and help – but the opposite is true of systems in which there is predominantly male only care (e.g., some fish). I guess amphibians are interesting in this context as they have both external and internal fertilisation thus presumably both male and female care can evolve. I wonder if this should be incorporated into the analysis?

Page 7, Line 135: Is origin and diversification a better term here than origin and maintenance?

Page 7, lines 139 – 141: Is this not expected given that attendance is necessarily a precursor to the other forms of care – that is that attendance must also be present in the context of the other care types. Perhaps this is a semantic question. But to me this question is about whether egg attendance evolves more readily than egg attendance with brooding/feeding/transport or is the definition of egg attendance separate to those other care types? To me this is not at all surprising and it needs to be coaxed in with the other results to really maximise its interest. That attendance is a precursor to more complex forms of parental care and that parental care in eggs is a precursor (or not) to parental care at other juvenile life stages. The fact that attendance is more likely to be lost also makes sense in this context, but I am not sure it is the big sell of this paper.

Page 7, Lines 141 – 145: I was confused by this statement. The statement here suggests that the rate of loss of tadpole feeding is high (“being high for traits such as tadpole feeding”) – which is supported by the data in figure 1c and table S2 – but this then appears at odds with the data presented in figure 1d which suggested that there is no evidence of an evolutionary loss of tadpole feeding. The opposite is true for egg attendance. Am I missing something here?

Page 8, Lines 152 – 154: How relevant is it to collapse the data across functional context. While I can appreciate that for a simple trait such as attendance is equivalent – it's a simple increase in association between parents and offspring – I am not sure whether this is the case for brooding and or transport. This is important because the interpretation here lies in the assumption that the complexity of these behaviours is equivalent across life stages – which might not be the case – egg brooding or transport for example may be less complex than juvenile transport.

Page 9 and 10, Lines 191 – 192: Here and throughout you report the Log BF outputs of your models but for someone not familiar with these models it is difficult to know what a particular LogBF value means in terms of effect size. Perhaps this can be made clearer.

Page 9 and 10: Lines 190 – 194: These two sentences appear to directly contradict one another. “We find strong support for correlated evolution between early and late care. Parental care at the egg stage is equally likely to evolve and revert to no care, as it is to facilitate the evolution of prolonged care through the acquisition of care at the later development stage.” This is then further contradicted in Lines 199 – 201 which states that nearly all the evolutionary origins of late care have evolved when early care is already present”. Perhaps I have missed something, but I think these statements need to be cleared up.

Page 9 and 10: Lines 190 – 194: Does the data also suggest that late care is also just as likely to facilitate the evolution of care across both early and late stages (figure 2)? This does not seem to be discussed anywhere. Is this a likely scenario in reality?

Page 10, Lines 194 – 196: See comment above but I think this is a neat result that will provide a foundation for future work. It suggests that alternative pressures may influence care at different life stages and thus it would be interesting to know what they are. As suggested above, there is some precedence for this for other social traits.

Page 10, Lines 201 – 203: Ok but would you expect that tadpole feeding per se would be likely to have an evolutionary link to parental care at the egg stage beyond the fact that parental care at the egg stage may mean that there is more likely to be care at the larval/juvenile stage full stop? As detailed above, this suggests to me that you are more broadly interested in the extent to which any form of care at the juvenile stage is likely to have been preceded by care at a previous stage. The elaboration of care within each stage is then a separate question.

Page 11, Lines 226 – 234: I do not see how this information follows from the previous information regarding the evolution of male, female or bi-parental care. Indeed, as you have no information on the extent to which the evolution of increase parent-offspring associations and elaboration of care then is mediated by, or mediates, levels of conflict within the brood I would suggest dropping these discussion points.

Page 12, Lines 240 – 243: As detailed above, it is not clear what you mean by a reduction in the duration of care? Does this mean that species can go from caring right across the offspring's developmental period to only one or two stages? If so, I am not sure I can see where that specific data is reported. What your data reports is that parental care is just as likely to be last as it is to result in more prolonged care (e.g., Figure 2). I think loss of care as opposed to reduced duration I perhaps a more appropriate word here.

Page 12, Lines 255 – 257: Again, I am not sure what you mean here by the correlated evolution between male and female care. Does this mean that where different care traits evolve they are as equally likely to evolve in males compared to females? This seems interesting to me – as detailed above, would we not predict that male and female only care would evolve under different scenarios based on, for example, fertilisation mode which I presume would have a strong phylogenetic signal. The fact that both male only and female only female care evolves under the same phylogenetic circumstances makes me wonder what are the circumstances under which one evolves vs the other. Is it worth discussing this point?

Page 12 and 13, Lines 260 – 261: Is this not dependent on the extent of variation in fertilisation mode which has been suggested to mediate the outcome of sex specific conflict in care? In systems that include both male only and female only care then bi-parental care has a greater opportunity to evolve from a background of both. This is in contrast to various other terrestrial vertebrates – for example mammals and birds – in which females are the primary care givers and therefore the most parsimonious (and in many instances only) route to bi-parental care is males joining females. These points I think need to be discussed (see also previous point).

Page 13, Lines 261 – 263: Higher rates than what?

Page 13, Lines 263 – 267: It would be nice to have an idea of the number of transitions to and from male/female only care and biparental care within these traits so that the reader can interpret the data in the context of the statistical power. I could not find the data relating to this anywhere.

Page 13, Lines 279 – 283: Same goes for the data on complementary vs different care types. I also wondered how this works? How do you get a system in which males and females perform care behaviours at different stages of offspring development? This would require one of the sexes to remain in the other sexes home range and not perform any care at one stage but then take over at a later stage. What evidence is there that this occurs. Presumably, this must co-occur with other aspects of the mating social system.

Page 16, Lines 330 – 333: You state that data from reviews were supplemented with data from the primary and secondary literature but don't supply any information on how (I also could not find this in the supp info). What searches were done to complement those studies, what were the search terms used, what was the depth of literature searched?

Supp Info, Page 1, Lines 17 – 21: You state that you did not include short term egg attendance (egg attendance immediately after birth) but what about egg attendance that did not go throughout the entire period. This is important because egg attendance for a prolonged period at the end of the developmental period is likely to exert a much more significant effect on the likelihood of post hatching care than egg attendance for a prolonged period of time at the begging. What data is there out there on duration of attendance and when that attendance occurs?

Reviewer #3:

Remarks to the Author:

This manuscript describes broad-scale analyses of parental care behaviours in Amphibians. The breadth of the study is impressive and represents probably the most extensive data set of parental care for this taxon – this is particularly impressive given the wide range of care behaviours found in Amphibians. The analyses assess rates and patterns of transitions among care states and categories and show that simpler behaviours, such as egg attendance, evolve more rapidly than more complex care traits, such as viviparity. The analyses also show that biparental care is likely to be an unstable state, with transitions away from biparental care arising at a much higher rate than transitions towards biparental care. The manuscript addresses questions of broad interest in a system that is particularly important for understanding the evolution of parental care. I do have some queries and suggestions but I do not anticipate that substantial new analyses are necessary (but see suggestions in point 2 and potentially point 4 below).

1) Viviparity as a subdivision of care does not seem well justified. I would argue that it is a life history trait that can result in different forms of offspring demands and alters the selection pressures on care behaviours, rather than being a care behaviour itself. This distinction might be helpful in terms of explaining differences in gains/losses. Perhaps behavioural traits are more labile than life history or physiological traits because changes in traits such as viviparity may require more complex developmental/genetic changes. Perhaps a more useful/justifiable division would be behavioural versus physiological (or anatomical) transitions (see lines 15-18 on page 4).

2) Biparental care is very rare (17 species with complementary care and 14 with overlapping care) and I wonder whether this is problematic for the models. It is hard to tell from the figure, but it is not clear that any losses are actually observed so it is not clear where the inference of high rates of loss comes from and I was left wondering if the result is reliable or whether the model assumptions have been violated. With the inferred rate of loss being so much higher than the rate of gain of biparental care, it seems remarkable that biparental is observed at all. The challenge of inferring gains and losses of binary traits have been discussed and in particular the potential importance of differential speciation and extinction has been highlighted (e.g. Goldberg, E.E. and B. Igić. 2008. On phylogenetic tests of irreversible evolution. *Evolution* 62:2727-2741; Goldberg, E.E., J.R. Kohn, R. Lande, K.A. Robertson, S.A. Smith, B. Igić. 2010. Species selection maintains self-incompatibility. *Science* 330:493-495). Perhaps biparental confers different speciation or extinction rates compared to uniparental care and this could explain the distribution of care. It may not be necessary to re-analyse the data using such models but at the least some consideration of these alternative mechanisms is needed. Although plausible, I don't think that the results are convincing in terms of stability of biparental care, as opposed to care simply being rare. Perhaps a relatively straightforward way to

assess the reliability of the results would be to simulate a multistate character evolving on the tree under the estimated transition rate parameters and assessing the simulated frequency of character states compared to the observed – this could be achieved with the `sim.character` function in the R package `diversitree`. In addition, it would be helpful to see in more detail the distribution of biparental care on the tree. These details are in figure 4, panel a but it is hard to read. It either needs to be much bigger, or perhaps replotted to show more clearly where overlapping and complementary biparental care evolves (i.e. don't plot the grey circles and instead state that nodes lacking a symbol have no biparental care).

3) The raw data do not seem to be available – will they be deposited to e.g. Dryad on publication?

4) Losses in the table in figure 1 don't seem to be consistent with the transition rates, most obviously the rate of loss for tadpole feeding is clearly non-zero in panel c but the number of losses is estimated to be zero. The patterns for gains seem consistent (high rate associated with high numbers of gains), but losses don't appear to tally at all. I may have missed something but I wonder if there is an error in reporting of these results.

Additional minor comments:

It would be helpful to report a range or HPDs for transition rates and number of gains losses e.g. page 8, line 2, and in the supplementary tables (a range would be more helpful than reporting mean, median and mode).

Page 3 line 6: "Not only does parental care increase the fitness of offspring and parents": Is this true? If care increases fitness of parent and offspring then there should not be intrafamilial conflict. Isn't the point that care can effect fitness of individuals (offspring, mother, father) differentially and it is this variation that cause conflict?

Page 5, lines 10-12: This is a quite hard to follow (what does "parental care forms of diverse complexity" mean?).

Methods, page 17, line 14. The description of tree scaling is ambiguous – how was the tree scaled and why? Also, why use this rather than the more recent Pyron and Jetz amphibian tree?

Figure 1 - Is grey absence or lack of data? The figure is hard to read because of the small size.

Signed
Gavin Thomas

Please find below our point by point response to each of the reviewers' comments.

Reviewer #1 (Remarks to the Author):

Overall, I found this to be an important and interesting study well-worth publishing. While other studies of the evolution of parental care and reproductive strategies in amphibians have been done, this is the most extensive one that I know of. The authors have accumulated an impressive dataset that will, I think, be useful to comparative evolutionary biologists for years to come. Beyond this, the authors have been able to focus on what (in my opinion) are interesting questions that have not been adequately addressed in a comparative framework. The idea that simple characters should evolve more rapidly and be more volatile than complex characters is a concept of general interest in evolutionary biology, and the authors are able to address this issue in a convincing manner with their dataset. The stability of biparental care is another issue of general interest to both theoreticians and empiricists, and the authors of this study are able to provide some novel insights in this regard. The methods used are current, and appear appropriate to the questions they were used to address. I had just a few minor comments that should be addressed before publication, as noted below.

Thank you for these positive comments on our manuscript.

Line 67: The claim that behaviors that cannot easily classified as simple or complex should evolve at intermediate rates seems somewhat vague. I think it might be worth specifying exactly which behaviors those are, and making a specific case for each one.

***Response 1.** We agree that we have not made this point as clear as we wished and rephrased this section accordingly, adding a new paragraph (please see changes from Page 6, Lines 11 to Page 7, Line 2). The literature is clear that egg attendance is expected to be the simplest form of parental care and that viviparity is a complex care adaptation. However, we found no mention anywhere in the literature of how we should consider the many other forms of care that we find in nature with respect to complexity. Therefore, it remains unclear whether we should class care traits as either simple or complex, or if there is a continuum between the two extremes of egg attendance and viviparity and, if the latter, how we should rank care behaviours and adaptations along a 'complexity axis'.*

Care forms in amphibians are very diverse and traits such as feeding, transport and brooding exhibit some level of behavioural, physiological and morphological coadaptations between parents and offspring. We suggest that it is more likely that complexity is a continuum and propose that traits that are not explicitly very simple or very complex should be regarded as 'intermediate', although we do not attempt to any a-priori ranking of complexity given the lack of detailed anatomical, physiological and behavioural information on many care forms. We thus expect that these 'intermediate' care forms will probably exhibit rates of gain and loss that are somewhere between those of egg attendance and viviparity, and test if this is the case.

Line 187: I would be curious to know if the results remain (qualitatively) the same if behaviors (like viviparity) that span early to late stages were classified as later (I think a case could be made that this would be a logical category for that type of behavior).

Response 2. *We are unclear what the reviewer means here. We think that perhaps the reviewer is asking whether our results change if care forms such as viviparity are classified as late care only (instead of both early and late care, as we have done) – if this is not what the reviewer means, please ignore our explanation of our scoring system that follows below.*

We think that our current classification system is justified because with viviparity and egg brooding in direct developing species care is provided continuously from fertilization up to the completion of metamorphosis, and so these care forms should be scored as both early (egg stage in our scheme) and late care (tadpole and juvenile stage which are combined in our scheme). In other words, it wouldn't make biological sense to score these care forms as late only care, because care occurs also at the egg stage. Furthermore, classing them as late only care would be inconsistent with the classification for species in which parents do indeed provide care only at the tadpole stage (tadpole feeding) without any form of care at the egg stage. To clarify this, we have now added more details on viviparity and care forms with direct development (P20.L15-17; Supplementary Information P4.L8-10).

Line 370: Change "allows to reduce model complexity and over-parametrization" to "enables a reduction of model complexity and over-parametrization"

We have changed this as per your suggestion (P22.L8-11).

Reviewer #2 (Remarks to the Author):

In this paper Furness and Capellini report on evidence of parental care across over 1000 species of amphibians about 40% of which have been recorded as displaying some form of parental care. They use this data set to carry out several phylogenetic comparative analyses to elucidate the origins and evolutionary diversification of parental care. Specifically, they focus on three main hypotheses, a) that simple forms of parental care are more evolutionary labile than complex forms of parental care, b) that care during the early stages of juvenile development is a precursor for care at later stages of juvenile development and b) that biparental care will exhibit greater evolutionary stability when parents vary in what form of care they exhibit. They find some support for the first two hypotheses but limited support for the last hypothesis.

Overall this is an impressive data set that has the potential to contribute to our growing understanding of the origins and evolution of parental care. Amphibians are particularly tractable systems for exploring these questions because of the considerable diversity of parental care exhibited (functionally, taxonomically and in terms of which sex provides the care) providing opportunities to develop and test clear hypotheses above the factors responsible for that variation. The paper goes some way to addressing some of these questions and will no doubt provide a basis from which future hypothesis driven tests of the mechanisms underpinning some of the patterns elucidated could be undertaken. However, I had several questions regarding the conceptual basis of the paper that I think need to be addressed before the paper can really

deliver on its promise. Furthermore, there are several key aspects of the analysis that need to be addressed before the results can be interpreted with confidence. Below I detail these concerns.

Thank you for the detailed comments and suggestions. Below we address each of your concerns.

Major Comments

My major comment regards the overall conceptual framework from the paper with respect to the specific aim of understanding patterns underlying the origins and evolution of parental care. This is a substantial challenge and one which the amphibian data set is potentially well established to test – however, I am not sure the framework of your paper is really set up appropriately to explore this question in the most powerful way possible. The paper at the moment is set up to explore what read at time as three disparate aims that are of relevance, broadly, to understanding the evolution of parental care. These aims are all interesting and important to address, but I think they probably want to be brought together so that the paper has a single narrative to it. At the moment it appears, in places, that the paper is there to tick off a number of understudied areas in parental care research rather than to use the data to understand evolutionary patterns of parental care in amphibians in a holistic manner. To address this I think that the paper could perhaps be split up into three sections – the first is understanding the origins and elaboration of care across different life history stages, the second is understanding the elaboration of care across functional context within life history stages, and the third is a question about the extent to which these differ between males and females and the evolution of bi-parental care more broadly. I think this would allow you to provide a more powerful and structured analysis of the evolutionary patterns of parental care diversification than you currently report. Indeed, at present the initial major focus is on the evolutionary lability of simple vs complex traits. This is of course relevant to the data on parental care that has been collected for this paper but is perhaps represents a more general question than the one being posited here. Instead I think that this aspect of the paper would be better incorporated into a more explicit focus on parental care itself where the results in terms of variation in the evolutionary lability of traits emerges as an outcome of the models rather than as an explicit focus.

***Response 3.** We acknowledge that we could have better united the three related research questions that we address in our manuscript – on type of care and complexity, care duration and stage, and caring sex and biparental care. To this end, we have substantially revised the text throughout to improve the way we integrate our three major themes (e.g. P4.L19-22; P5.L4-5, P7.L18-20; P8.L9-13; P12.L10-11; P13.L6-11; P14.L9-12; P15.L22-23; P18.L13-18). Indeed, we thank the reviewer for the other suggestions (below) to look at which behaviours at a given stage promote the evolution of different care forms at subsequent stages. This has helped us to better meet our overarching goal and show that, if we are to understand how parental care evolves, we do need to fully consider the diversity of parental care in its entirety, rather than classing parental care as a rough present/absent trait. To our knowledge, our study is the first to do so and this has allowed us to unravel unexpected and unique set of results that we think will help drive this research field forward.*

With regard to the structure of the ms, we have retained the overarching presentation of our original submission because it is not fundamentally different from the one proposed by the reviewer. To address the reviewer's concern, though, we have revised the text to improve clarity throughout.

To address these concerns, I would suggest a slight change in overall focus. I think the major focus should be on the extent to which simple forms of early parental care (care at the egg stage, viviparity) sets the stage for prolonged parental care at the larval and juvenile stage. This is the key question that your data allows you to address – given that you do not have predictors for what drives the initial emergence of care itself. When doing this, I would also suggest simplifying this analysis somewhat.

***Response 4.** We fully agree with the reviewer that a key question is whether care at an earlier stage is essential for the evolution of care at later stages. This is precisely what we have tested in our study. Specifically, we have achieved this by investigating whether early care promotes the acquisition of care at*

the tadpole and juvenile stages, and thus leads to the evolution of prolonged care (i.e. when both early and late care are present; please see P12.L8-P15.L21). We think that this confusion has arisen by some misunderstanding of our analytical framework and how this allows us to test the question(s) asked in our study. To address this, we have revised the text and explained more in detail how the hypotheses translate into testable predictions that can be assessed using our methods (please see P7.L16-18; P12.L16-P13.L2; P15.L23-P16.L4; P17.L14-16; P23.L1-20; P24.L7-12). We have also included a Supplementary figure that summarises the predictions in relation to our analytical framework (Figure S1 in SI).

Briefly, we use Discrete models of evolution to evaluate whether the evolution of early care (egg stage) from no care promotes the evolution of late care, thus leading to prolonged parental care (when both early and late care are present). Discrete requires 2 binary (i.e. absence/presence) variables, e.g. 'early' and 'late' care. First, we need to evaluate whether these two variables are in fact correlated across the phylogeny by comparing the Discrete Independent model (in which the 2 variables are evolutionary unrelated) vs the Discrete Dependent model (in which they are evolutionarily associated). Next, if the Dependent model provides a better fit to the data than the Independent model (Log Bayes Factor greater than 2), the 2 variables of interest have evolved in a correlated fashion. In this case, we look at the magnitude of the transition rates across the 4 combinations of character states that the 2 binary traits can jointly take to identify evolutionary pathways.

Note that amphibians may have no care (n=912); early care only (i.e. egg stage, n=144); late care only (n=8, i.e. species with tadpole feeding that exhibit no early care at all), or both early and late, i.e. prolonged (n=258, including species with egg attendance and tadpole care behaviour, plus species with viviparity, egg attendance or egg brooding with direct development; please see Figure 3B and 1A). Thus, our analytical framework is well suited to analyse these data.

For the hypothesis to be supported we should thus find that (i) early and late care are evolutionarily 'correlated' (the Dependent model fits the data better than the Independent model) and (ii) the transition from 'early care only' to 'early and late care (i.e. evolution of prolonged care)' is greater than zero (please see Figure S1a in Supplementary Information).

Note that our analytical framework also allows us to evaluate whether a particular combination of character states is evolutionarily stable by considering the magnitude of transition rates 'into' vs 'out of' it. Specifically, if prolonged care (early plus late) is evolutionarily stable, the transition rates leading into it should be higher than those leading to its loss (i.e. towards short term care only). Our analysis shows that early care and late care are indeed evolutionarily associated, that early care promotes the evolution of prolonged care, but prolonged care can be as easily lost as it is gained (P13.L3-6, Figure 3). This indicates that prolonged care evolves in 2 steps, first through the evolution of early care then by acquiring late care, but once evolved prolonged care can be reduced to short term care. We show that this is irrespective of caring sex as we find qualitatively similar results in both females and males (P13.L6-11).

We use the same analytical framework for the evolution of biparental care (P15.L22-P17.L7, P23.L10-P24.L12, Figure S1b; please see also response 22) and for the additional analyses requested by Reviewer 2 (P13.L12-P14.L8; please see responses 5 and 7).

That is, I would first simply ask whether care at the larval/juvenile stage is more likely to evolve from a background of egg care/viviparity or no care. To achieve this, I wonder if you need the diversity in care types for this initial step and instead just include the presence and absence of care at each of those stages. Instead you could focus on a simple framework like this (apologies if this is slightly oversimplified):

- a) Egg Care --- larval Care --- Juvenile Care
- b) Egg Care (with direct development --- Juvenile Care
- c) Viviparity --- Juvenile Care

You would then be interested in asking very simple questions regarding the extent to which Juvenile care evolves as a function of any earlier attendance (e.g., combine a, b and c) or whether this differs for the different early stage care backgrounds (separate analyses for a, b and c). This would then allow you to not only ask whether parental care at the juvenile stage evolves as a function of parental care at earlier stages but also whether the context of that early care matters (similar to the recent analysis done for reptiles). You could also do a smaller separate analysis for the extent to which egg care predicts larval care.

Response 5. *We thank the Reviewer for these suggestions; we have now included these analyses in the revised ms but have done so by considering individual behaviours where sample sizes are sufficiently large for the analyses (P13.L12-P14.L8 and Tables S5 in Supplementary Information). The analyses in pathways (b) and (c) show that egg attendance with direct development – but not egg brooding with direct development – is evolutionarily associated with and promotes the acquisition of care at the Juvenile stage. However, we found no evidence that viviparity promotes the evolution of juvenile care as our analysis reveals that these two traits are not associated.*

Please note that to investigate the proposed pathway (a) we need to divide this pathway into its 2 elements and run 2 separate analyses, as Discrete can only work with 2 binary variables at a time (please see response 4 above). The first part of this analysis (Egg care to Tadpole care) is identical to our analysis of Early care vs Late care in Figure 3 and Table S4 (where our Late care variable combines tadpole and juvenile stages), since all species that exhibit juvenile care also have tadpole care. We have thus studied how individual egg care behaviours promote the evolution of which care behaviour at the tadpole stage, where sample sizes are sufficiently large. We find that egg attendance without direct development promotes the evolution of tadpole attendance and tadpole transport but is unrelated to tadpole feeding. For the second part – tadpole care to juvenile care – the analysis fails to converge, therefore we do not present this analysis in the revised ms. The poorer performance of this latter analysis in MCMC is likely due to the rarity of juvenile care in Amphibians (n=23 species across all forms; please see Figure 1a) and the fact that juvenile care is found primarily in the Gymnophiona (caecilians); moreover 0 species have juvenile only care. These issues thus prevent us from investigating further whether tadpole care promotes the evolution of juvenile care in amphibians.

Because of the phylogenetic distribution and rarity of juvenile care in amphibians, we also do not want to emphasize too much juvenile care, which can be better studied in other taxonomic groups – like reptiles – where it is more common. Conversely, the real value of using amphibians in a study of parental care evolution like ours lies in the very diverse nature and richness of care forms at the egg and tadpole stages (please see Figure 1a) on which we wish to retain the attention of the study.

To sum up, we have included additional analyses of how individual care behaviours at one stage promote the evolution of care behaviours at subsequent stages, where sample sizes are sufficiently large. These new analyses complement but not replace our original analysis as the latter provides generality of conclusions and, by combining all care forms in a given stage, it is based on larger sample sizes.

Combined I think this would provide a more robust analysis of how parental care evolved and the extent to which care in the later stages is dependent on care in the early stages (and how that differs depending on the nature of that early care). Interestingly, as your data suggests, it would also allow you to compare this to instances where the evolution of care in early vs late stages represent independent evolutionary events. This would provide a nice framework for a discussion (and potentially future analyses) on the factors that might have promoted this (e.g., there is different selection on care at the different stages). There are some analogous systems in which these questions have been studied. For example, Michael Griesser's work on the evolution of family living and cooperation in birds (PlosBiol 2017).

Response 6. *We thank the reviewer for flagging this ms which we now cite. We find that early care promotes and is associated with late care in amphibians (P13.L3-P14.L8, Table S4 & S5). Thus, we find clear evidence that care at the later stages of development care is not evolutionarily independent from care at the earlier stage in amphibians and so it is more likely that selection at different developmental stages is*

similar. The only exceptions to this concerns tadpole feeding and egg attendance, and viviparity and juvenile care (P13.L15-P14.L8); we discuss particularly how the peculiar ecological conditions in which tadpole feeding is found may have made care at the egg stage unnecessary (P13.L19-P14.L3). Please see also response 7.

Importantly, the above approach would also then allow you to explore the question of the extent to which parental care elaborates within each of these care types. That is, do more complex forms of parental care evolve from simple forms of care. So, within these care types, does egg attendance provide a precursor for the evolution of brooding, feeding, transport or do those traits emerge independently (although see requirement for clarification on this point below). In this context questions about the lability of simple vs complex traits perhaps becomes more relevant as it allows you to say something about the evolutionary lability of simple attendance across these different stages and then how this has provided the basis from the evolution of further elaboration of care - and how that subsequently influences the maintenance of care.

Re-structuring the analysis in this way would also help clarify the role that traits such as viviparity play in this process. The complication is that viviparity can be viewed as both an evolved parental care strategy as well as a fundamental precursor to greater elaboration of care types. You seem to focus largely on the former in this instance (e.g., you state that viviparity is an extended form of parental care to occurs across life stages) but the above would allow you to use viviparity more powerfully to explore some of the precursors to extended forms of parental care. Indeed, one of the benefits of amphibians is that they exhibit multiple parity forms that provide fundamentally different associations between mothers and their offspring – which may have implications for the subsequent evolution of prolonged care (as well as other traits – as articulated in the introduction).

Response 7. *We thank the reviewer for these helpful suggestions; we have now included additional analyses in our study on how early care behaviour promotes the evolution of later care (P13.L12-P14.L8, Table S5; please see also responses 5 and 6). Specifically, in the revised ms we now show that egg attendance without direct development promotes the evolution of both tadpole attendance and tadpole transport (but not tadpole feeding), and egg attendance with direct development – but not egg brooding with direct development - promotes the evolution of juvenile care. However, viviparity does not promote care at the juvenile stage (P14.L4-8). These results complement and underpin our broader analyses that care at an earlier stage promote care at the later stages, thus early care promotes the evolution of prolonged care. However, they also reveal that the evolution of prolonged care is not strictly due to the evolution of more complex traits since, for example, attendance at the egg stage promotes attendance at the tadpole stage. By bringing together analyses on trait complexity (P9.L12-P12.L7) with those on the evolution of prolonged care (P12.L8-P15.L21), we thus propose that the evolutionary increase and the reduction in the duration of care depends on the care forms and how it is affected by parent-offspring conflict and sibling competition (P14.L9-P15.L4; P15.L14-21; P18.L12-21).*

Minor Points

Introduction

Page 3, Lines 42: Minor point but I would finish the first paragraph here and start a second paragraph from “While care behaviours...”. I think this would provide a more powerful introduction to the paper.

In this first paragraph we aim to provide a short overview of the field, and contrast what is known with what is not. As such we feel that the two sections are strongly linked and better kept together within the same paragraph.

Page 3, Lines 43 – 46: This is a long sentence I would consider splitting it in two.

We have now revised this sentence as suggested (P4.L11-15).

Page 3, Lines 43 – 46: I would also argue that the main reason for this is that in few systems is there the requisite diversity of care types to allow for a proper analysis of these traits.

Response 8. *We agree with the reviewer that amphibians (as well as fish and insects, at least) show an astonishing diversity of care forms; however, there is still great diversity in care adaptations in many groups that is often ignored. For example, we previously showed that accounting for such diversity, even in groups less diverse than the amphibians, can lead to major step forwards in our understanding of parental care evolution and how different care behaviours relate to the species' reproductive strategies (West and Capellini 2016 Nature Communications).*

Page 5, Lines 80 – 83: Is there data on these traits in offspring that could be tested in the context of the predictors of the loss of different care types? This may be outside the scope of the study, but it seems to me that this data set may also allow you to explore the extent to which the evolution of different care types have resulted in the evolution of offspring traits (perhaps focussing on the evolution of increasingly altricial offspring) – which may help explain the lability of different care types.

Response 9. *We agree that this is an interesting question but data on degree on altriciality, such as tadpole size or age when leaving the parents, are simply not available for the vast majority of amphibian species and are extremely plastic in amphibians, making testing such a suggestion extremely challenging to investigate at large comparative scale. We also agree that this question is outside the scope of our ms.*

Page 6, Lines 105 – 106: Does this not depend on the sex that has predominantly evolved to care for the offspring? For example, in systems where there is predominantly female only care bi-parental care occurs when males also stay and help – but the opposite is true of systems in which there is predominantly male only care (e.g., some fish). I guess amphibians are interesting in this context as they have both external and internal fertilisation thus presumably both male and female care can evolve. I wonder if this should be incorporated into the analysis?

Response 10. *We fully agree that amphibians are an excellent system in which to investigate questions on the evolution of biparental care. As the reviewer points out, our data and analysis reveal that there is no predominant caring sex and evolutionary pathway leading to biparental care in amphibians. We show that in amphibians biparental care evolves from either female or male uniparental care, by males joining caring females or by females joining caring males (P16.L4-13; Figure 4). We also test whether the additional help provided has to be in a different form of care (i.e. division of labour, which we classify as complementary care) or not (overlapping care) to make biparental care an evolutionary stable state, i.e. a condition that once evolved is harder to lose. However, we find that biparental care is always lost rapidly regardless of whether it is overlapping or complementary, hence it is not an evolutionarily stable strategy (P17.L8-P18.L6).*

We do not test whether fertilization mode promotes the evolution of female care vs male care because in amphibians mode of fertilization is little variable, in that most Caudata and all Gymnophiona have internal fertilization while nearly all Anura have external fertilization; thus Amphibians are not the best group in which to test this hypothesis. Moreover, the question has already been answered by Beck (1998) who demonstrated with phylogenetic analysis that fertilization mode and female/male care are unrelated in amphibians (Animal Behaviour, 55: 439-449). The inclusion of this question is also beyond the scope of our ms.

Page 7, Line 135: Is origin and diversification a better term here than origin and maintenance?

Response 11. *Diversification is typically used in the context of speciation and we do not use it in our ms to avoid possible confusion. The word 'maintenance' is more suited to the aim of our ms since we investigate trait persistence over evolutionary time (i.e. rate of loss compared to the rate of gain) through which we can*

infer, for example, whether prolonged care or biparental care are evolutionarily stable states. We acknowledge, though, that the word 'maintenance' may not be as clear as we wanted – to address this, we have now replaced it with 'evolutionary [origin and] persistence' in the sentence highlighted by the Reviewer (P9.L10-12) and similar wording across the ms.

Page 7, lines 139 – 141: Is this not expected given that attendance is necessarily a precursor to the other forms of care – that is that attendance must also be present in the context of the other care types. Perhaps this is a semantic question. But to me this question is about whether egg attendance evolves more readily than egg attendance with brooding/feeding/transport or is the definition of egg attendance separate to those other care types? To me this is not at all surprising and it needs to be coaxed in with the other results to really maximise its interest. That attendance is a precursor to more complex forms of parental care and that parental care in eggs is a precursor (or not) to parental care at other juvenile life stages. The fact that attendance is more likely to be lost also makes sense in this context, but I am not sure it is the big sell of this paper.

Response 12. *Please note that while egg attendance is often present with other forms of care, this is not always the case – egg attendance never occurs with egg brooding (with or without direct development) and viviparity, and is not be present in about a third of the species exhibiting tadpole feeding (please see Fig. 1A). For this reason, we classify behaviours and adaptation separately and by stage, and retain this distinction in this first analysis on trait complexity and function. Following this reviewer's suggestion, we now test whether egg attendance promotes the evolution of other forms of care and show that egg attendance without direct development promotes the evolution of tadpole attendance and tadpole transport while egg attendance with direct development promote the evolution of juvenile care (P13.L12-P14.L8; please see also response 7).*

More broadly on the significance of our results, please consider that a trait may be common in extant species either because it is gained and lost many times across the phylogeny (as it is the case for egg attendance in our study), or because it has evolved a few times deep in the tree and it has never been lost and has thus been inherited by many descendent species. The rates of evolution (i.e. rate of gain and rate if loss) are expected to differ under these 2 scenarios and both could in theory explain why egg attendance is common. Our analysis not only shows that egg attendance is gained and lost rapidly, but it also allows us to compare the rates of evolution across many traits of differing degree of complexity and occurring at different stage. We thus use egg attendance and viviparity as 'reference' points against which to compare the results for the other traits and this aids the interpretation of the results. To our knowledge, ours is the first paper that shows that different forms of parental care, including egg attendance, are gained and lost at different rates across stages of development, functions, and trait complexity. We think this is a major advancement that will be of interest to many.

Page 7, Lines 141 – 145: I was confused by this statement. The statement here suggests that the rate of loss of tadpole feeding is high (“being high for traits such as tadpole feeding”) – which is supported by the data in figure 1c and table S2 – but this then appears at odds with the data presented in figure 1d which suggested that there is no evidence of an evolutionary loss of tadpole feeding. The opposite is true for egg attendance. Am I missing something here?

Response 13. *We acknowledge that we have confused our readers here. We used Bayesian RJ MCMC Multistate models for our analyses as these are currently the most reliable and state of the art methods for evolutionary studies of discrete variables, since they can better explore parameter space and identify all suitable model solutions (Currie & Meade (2014) in Modern phylogenetic comparative methods and their application in evolutionary biology, Springer). Because the program we used to run these models – BayesTraits – does not have graphical tools associated with it, in our earlier submission we used stochastic character mapping in maximum likelihood (i) to represent visually the evolutionary history of individual traits on the phylogeny (Supplementary Figure 1 in our earlier submission), and (ii) to provide an approximate number of gains and losses of each trait individually to help readers less familiar with the interpretation of transition rates (i.e. the rates of gain and loss). Note, therefore, that both the figure in our*

earlier submission (Figure S1) and the estimate of gains and losses derived from stochastic character mapping are only one realization of all possible model solutions that MCMC can instead provide (and indeed it is a solution far less supported than others). With this in mind, we note that for all traits in our analyses, except tadpole feeding, results for RJ Multistate models are reasonably well represented by figures derived from stochastic character mapping. For tadpole feeding, however, stochastic character mapping underestimates alternative yet suitable model solutions hence leading to number of gains and losses (former Fig. 1d) and an evolutionary history (former Figure S1) that do not match the more reliable results from MCMC RJ Multistate analysis.

MCMC RJ Multistate analysis for tadpole feeding show that, on average, this trait has been gained at a relative low rate (and a much lower rate than egg attendance) but it has been lost very rapidly once evolved. The combination of these two transition rates helps explain why it is rare in extant species.

To avoid confusion, we now only present results from MCMC RJ Multistate analysis and have removed estimates of the number gains and losses derived from stochastic character mapping and the graphical representation of the evolutionary history that we previously reported as Figure S1. We have instead included additional information on the comparison and interpretation of transition rates to help readers less familiar with our analytical approach (P6.L7-10; P9.L12-16).

Page 8, Lines 152 – 154: How relevant is it to collapse the data across functional context. While I can appreciate that for a simple trait such as attendance is equivalent – it's a simple increase in association between parents and offspring – I am not sure whether this is the case for brooding and or transport. This is important because the interpretation here lies in the assumption that the complexity of these behaviours is equivalent across life stages – which might not be the case – egg brooding or transport for example may be less complex than juvenile transport.

Response 14. We agree with the Reviewer that attendance is presumably no less or more complex at the egg than at the tadpole or the juvenile stage. The same however applies to the other adaptations and behaviours that are defined with very similar wording in the literature, on which we base our definition of each care form (e.g. McDiarmid 1978; Crump 1995, 1996; Lehtinen & Nussbaum 2003; Wells 2007; all cited in our ms; please see also Supplementary Information; Definitions of care forms). Specifically, (i) with transport, both tadpoles and juveniles are carried by parents from one location to another; (ii) feeding at the tadpole and juvenile stage both require nutritional support by the mother through eggs (Anura) or sloughed-off skin (caecilians), and (iii) in both brooding at the egg and tadpole stage offspring develop on or within the parent's body. Because the description of these care forms in the amphibian literature suggests that they are functionally equivalent across stages of development, merging traits by function irrespective of developmental stage is justified. To help address the reviewer's point, we have provided further details on the definitions of care forms by stage in the Supplementary Information (SI P.1-4) and explained the details on why functional grouping are sound in the main text of the revised ms (P10.L3-12).

We decided to repeat the analysis by grouping traits by function regardless of developmental stage because functionally equivalent care forms are more common at earlier than later stages (Figure 1A, Supplementary Table 1). By considering functional categories irrespective of developmental stage we can therefore explore further the idea that trait complexity affects rates of evolution. We have now added additional text to better explain our reasoning and conclusions (P10.L3-12).

Page 9 and 10, Lines 191 – 192: Here and throughout you report the Log BF outputs of your models but for someone not familiar with these models it is difficult to know what a particular LogBF value means in terms of effect size. Perhaps this can be made clearer.

Response 15. Please note that Bayes Factors are not effect sizes, but an estimate of model fit to the data through the comparison of the marginal likelihoods of two alternative models. Thus, BF can be considered analogous to differences in maximum likelihood scores assessed through likelihood ratio test in ML framework (Kass R. E., Raftery A. E. 1995 Bayes Factors. Journal of the American Statistical Association 90,

773-795; Raftery 1996 Hypothesis testing and model selection, in Gilks et al Eds Markov chain Monte Carlo in practice, Chapman & Hall/CRC). We thus use BF to compare the fit to the data of alternative models of evolution, i.e. the Independent and the Dependent Discrete models in our analyses of early and late care (i.e. evolution of prolonged care), and of female and male care (evolution of biparental care). To address the reviewer's point, we now provide additional explanation of BF (P24.L2-7).

Page 9 and 10: Lines 190 – 194: These two sentences appear to directly contradict one another. “We find strong support for correlated evolution between early and late care. Parental care at the egg stage is equally likely to evolve and revert to no care, as it is to facilitate the evolution of prolonged care through the acquisition of care at the later development stage.” This is then further contradicted in Lines 199 – 201 which states that nearly all the evolutionary origins of late care have evolved when early care is already present”. Perhaps I have missed something, but I think these statements need to be cleared up.

Response 16. Without further details, we find it difficult to see how these statements are in contradiction and we confirm that they accurately reflect our results in the context of the Discrete analytical framework. We suspect that one reason for the Reviewer's comment might be that we need to better explain how the Discrete model of evolution help us test the hypotheses and predictions. To this end we have added further details on the Discrete framework and predictions (P12.L16-P13.L2), including a Supplementary figure (Figure S1a) on the predictions for this analysis (please see also Response 4). With regard to the first sentence here highlighted by the reviewer, what we show is that the magnitude of the transition rate from no care to early care is similar to the magnitude of the transition rate of its loss (i.e. early care back to no care) and it is also similar to the magnitude of the transition rate from early care to prolonged care (please see Figure 3B, Table S4a). These results indicate that early care can as easily evolve from no care as it can revert back to no care; and early care can also as easily lead to the acquisition of late care and so the evolution of prolonged care. In essence, early care is a dynamic state that evolves easily, can be lost as easily as it is gained, or it can as easily lead to an increase in care duration. We have reworded the sentence flagged by the reviewer to better describe these results (P13.L3-6). With revision of the text and inclusion of the additional analyses suggested by this reviewer (see response 4), the second statement flagged by the reviewer becomes unnecessary and has been deleted.

Page 9 and 10: Lines 190 – 194: Does the data also suggest that late care is also just as likely to facilitate the evolution of care across both early and late stages (figure 2)? This does not seem to be discussed anywhere. Is this a likely scenario in reality?

Response 17. Our overall analysis on the combined care forms (including traits such as brooding, viviparity and feeding) suggests that late care alone (i.e. in the absence of early care) is an evolutionarily unstable condition i.e. once evolved it is quickly lost either to no care or to prolonged care through the acquisition of early care (P13.L3-6; Figure 3, Table S4). The additional analyses, on individual care behaviours at the egg and tadpole stage with sufficient sample sizes, show that tadpole attendance and tadpole transport do not evolve in the absence of egg attendance (the transition rate from no care to tadpole only care is estimated as 0 in all models; Table 5a & b), while tadpole feeding evolves independently of early care (P13.L15-19). We discuss these results in the revised ms (P13.L19-P14.L3). Please see also response 6, 7 and 18.

Page 10, Lines 194 – 196: See comment above but I think this is a neat result that will provide a foundation for future work. It suggests that alternative pressures may influence care at different life stages and thus it would be interesting to know what they are. As suggested above, there is some precedence for this for other social traits.

Response 18. We agree that, in principle, different ecological pressures might select for different care forms at different stages. However, we find that early and late care are evolutionarily associated (P12.L15-16). Moreover, early care is a stepping stone for the evolution of prolonged care in general and of the behaviours tested across stages (e.g. tadpole attendance, tadpole transport, juvenile care; P13.L3-P14.L8, Table S4, Table S5). We found this is not the case for egg attendance and tadpole feeding, and viviparity and

juvenile care and we discuss these results in the revised ms (P13.L19-P14.L8). Please see also response 6, 7 and response 19 below.

Page 10, Lines 201 – 203: Ok but would you expect that tadpole feeding per se would be likely to have an evolutionary link to parental care at the egg stage beyond the fact that parental care at the egg stage may mean that there is more likely to be care at the larval/juvenile stage full stop? As detailed above, this suggests to me that you are more broadly interested in the extent to which any form of care at the juvenile stage is likely to have been preceded by care at a previous stage. The elaboration of care within each stage is then a separate question.

Response 19. *The sentence the reviewer is referring to here is no longer present in the revised ms (see Response 13). Following this reviewer's earlier suggestion (see response 7), we now demonstrate that, unlike tadpole attendance and tadpole transport, the evolution tadpole feeding is unrelated to the evolution of egg attendance (P13.L15-19). We retain the discussion on the peculiar ecological conditions in which tadpole feeding is found (P13.L19-P14.L3) and how this explains why species with tadpole feeding do not always have egg attendance (Figure 1a).*

Page 11, Lines 226 – 234: I do not see how this information follows from the previous information regarding the evolution of male, female or bi-parental care. Indeed, as you have no information on the extent to which the evolution of increase parent-offspring associations and elaboration of care then is mediated by, or mediates, levels of conflict within the brood I would suggest dropping these discussion points.

Response 20. *We have restructured the presentation of the results so that the analyses by sex is presented earlier in the ms and better integrated with the narrative of the other results (P13.L6-11). In the rest of paragraph (P14.L9-P15.L4) we now only discuss results on the evolution of prolonged care from early care through the acquisition of late care, and given our results, we ask how and why parental care can increase in duration. Based on the literature we identify parent-offspring conflict and sibling competition as two important mechanisms. Specifically, we suggest that increased care duration is facilitated when the offspring can influence care levels, e.g. through begging and hormonal control of parental physiology (as also proposed by others, e.g. Gardner & Smiseth 2010 Proc B; Royle et al 2002 TREE; Haig 1993 Q Rev Biol; all cited in our ms). We acknowledge that we have not explained well enough how begging and hormones are relevant in amphibians. To address this, we now provide more details on solicitation displays in tadpoles that are analogous to begging, and on how under such circumstances parent-offspring conflict and sibling competition are expected (P14.L15-19). Moreover, in amphibians both sexes can brood offspring on or inside their bodies (i.e. in the back, stomach, vocal sac, as well as the oviduct) and therefore we suggest that there is great potential for offspring to influence care levels through the release of hormones that manipulate parental resource allocation (P14.L19-P15.L4). We believe that our results will stimulate important research in brooding in amphibians and lead to fundamental insight, beyond the analogous well studied cases in placental mammals, on how offspring manipulate parents and care levels in the context of parent-offspring conflict.*

Page 12, Lines 240 – 243: As detailed above, it is not clear what you mean by a reduction in the duration of care? Does this mean that species can go from caring right across the offspring's developmental period to only one or two stages? If so, I am not sure I can see where that specific data is reported. What your data reports is that parental care is just as likely to be last as it is to result in more prolonged care (e.g., Figure 2). I think loss of care as opposed to reduced duration I perhaps a more appropriate word here.

Response 21. *Our analytical framework divides care duration in short term (early only, e.g. egg attendance, or late only, e.g. tadpole feeding) and prolonged care (when both early and late care are present); in addition, care may be absent altogether (see also Figure S1a in SI). Thus, in Figure 3A in the revised ms, species with prolonged care are those with both early (blue) and late (red) care. Parental care duration can thus be reduced (i) when prolonged care is reduced to short term care (egg stage only or tadpole stage only); and (ii) when short term care is lost altogether, i.e. back to a condition of no care. Results on the*

reduction of care are summarised in the diagram in Figure 3B depicting evolutionary pathway from prolonged care to short term care only (either early or late only) and from short term care to no care.

In the sentence flagged by the reviewer, a reduction of care duration refers to the loss of late care from prolonged (thus back to early only or late only care) as well as the loss of short term care altogether to no care, thus the phrase 'loss of care' suggested by the reviewer would not fully represent how care duration can be reduced. We think this confusion arises from our insufficient description of our analytical framework; we have therefore revised the text describing our approach more extensively (e.g please see P7.L16-18; P12.L16-P13.L2; P23.L1-20; P24.L7-12; Figure S1 in SI). Please see also responses 4 and 16.

Page 12, Lines 255 – 257: Again, I am not sure what you mean here by the correlated evolution between male and female care. Does this mean that where different care traits evolve they are as equally likely to evolve in males compared to females? This seems interesting to me – as detailed above, would we not predict that male and female only care would evolve under different scenarios based on, for example, fertilisation mode which I presume would have a strong phylogenetic signal. The fact that both male only and female only female care evolves under the same phylogenetic circumstances makes me wonder what are the circumstances under which one evolves vs the other. Is it worth discussing this point?

Response 22. *Biparental care arise when both males and females care. Our analytical framework asks whether the evolution of male and female care are independent of one another (Discrete Independent model) or the evolution of one is contingent on that of the other (Dependent model or model of correlated evolution). In the Independent model, male care and female care evolve in an unrelated fashion – i.e. whether males care or not is not influenced by what females do and vice versa; thus biparental care is simply the result of the sexes evolving care at the same time, irrespective of one another, in a given species. In the Dependent model, instead, the evolution (i.e. gain or loss) of care by one sex is contingent to that of the other sex; in other words, the evolution (gain or loss of) care in one sex influences the evolutionary trajectory of parental care in the other sex. The wording 'correlated evolution' refers to the comparison between the Independent and Dependent model of evolution through BF that shows that the Dependent model fits our data better. Thus, in the sentence flagged by the reviewer we indicate that we find stronger support for the Dependent model over the Independent model. We have however rephrased this sentence to avoid any confusion (P16.L4-8) and revised the text on the predictions and explanations of our analytical framework (P15.L23-P16.L4; Figure S1b; please see responses 4 and 10).*

In the Dependent model, biparental care may in principle arise either through a pathway where female uniparental care evolves first and males join females, or the opposite. In the Discrete Dependent model both pathways are modelled but the data may or may not find that they are equally possible. Our results show that both evolutionary pathways are indeed equally possible in amphibians; uniparental care by males and uniparental care by females are equally likely to evolve from no care; and biparental care can evolve by either sex joining (P16.L4-13, Figure 4B & C). With regard to mode of fertilization, please see response 10.

Page 12 and 13, Lines 260 – 261: Is this not dependent on the extent of variation in fertilisation mode which has been suggested to mediate the outcome of sex specific conflict in care? In systems that include both male only and female only care then bi-parental care has a greater opportunity to evolve from a background of both. This is in contrast to various other terrestrial vertebrates – for example mammals and birds – in which females are the primary care givers and therefore the most parsimonious (and in many instances only) route to bi-parental care is males joining females. These points I think need to be discussed (see also previous point).

Response 23. *Please see response 10 with regard to mode of fertilization which has already been shown to be unrelated to the evolution of parental care in amphibians. Previous studies show that indeed female uniparental care is the precursor of biparental care in mammals, birds and insects (Reynolds et al 2002 Phil Trans R Soc; Gilbert & Manica 2015 Evolution). However, we find that in amphibians biparental care evolves from a background of either male or female uniparental care (please see response 22) and we discuss how*

the sexes differ (or not) in the expression of care in the context of biparental care (P8.L17-19, P16.L15-P17.L7) and parental care more broadly (P13.L6-11; P15.L5-12).

Page 13, Lines 261 – 263: Higher rates than what?

Response 24. *In response to this and earlier comments which relate to our analytical framework in Discrete, we now clarify that we can compare the magnitude of transition rates ‘into’ or ‘out from’ a combination of character states and infer whether that combination is evolutionary stable (transition rates leading to the combination are higher than out of it) or not (transition rates out are higher than into it) (P15.L23-P16.L4; P16.L13-15; Figure S1b in SI). Please see also Response 4.*

Page 13, Lines 263 – 267: It would be nice to have an idea of the number of transitions to and from male/female only care and biparental care within these traits so that the reader can interpret the data in the context of the statistical power. I could not find the data relating to this anywhere.

Response 25. *Please see Figure 4A which shows the evolutionary history of female (red) and male (blue) uniparental care along the phylogeny, i.e. showing multiple independent gains and losses of both traits across the tree. This figure, combined with sample sizes for each combination of character states in Figure 4C (i.e. no care: N=912; uniparental male care N=158; uniparental female care N=221, biparental care N=31), provides the information on the power of the phylogenetic analysis. In our analytical framework, biparental care occurs when both female and male care are present and evolves in a contingent fashion from either uniparental female or male care. Discrete in BayesTraits however does not provide estimates of the number of gains and losses for single traits or combination of character states, but only the transition rates. However, please note that the number of gains and losses is an output of the model and depends on the transition rates, which we do report in both the Figures and Supplementary tables. Given the confusion we accidentally created in the original submission when presenting number of gains and losses derived from stochastic character mapping - a less powerful method than the more advanced Discrete models we use for analysis - we refrain from adding this information here; indeed we have deleted any details on number of gains and losses from the revised ms. Because MCMC RJ Discrete models are more powerful and much more informative than alternative methods we have instead provided additional text to better explain our analytical framework and additional details to aid the interpretation of the transition rates (please see P7.L16-18; P12.L16-P13.L2; P15.L23-P16.L4; P17.L14-16; P23.L1-20; P24.L7-12; please see response 13).*

Page 13, Lines 279 – 283: Same goes for the data on complementary vs different care types. I also wondered how this works? How do you get a system in which males and females perform care behaviours at different stages of offspring development? This would require one of the sexes to remain in the other sexes home range and not perform any care at one stage but then take over at a later stage. What evidence is there that this occurs. Presumably, this must co-occur with other aspects of the mating social system.

Response 26. *We define complementary biparental care as occurring when males and females perform distinct behaviours at the same developmental stage or the same behaviour but at different stages, or when at least some behaviours are unique to one sex at a given stage. Conversely, we define overlapping biparental care as cases in which both sexes undertake the same behaviour at the same stage of development (please see P17.L8-16, P21.L10-18). Our definitions are based on evidence in the literature from which we extracted the data and the data on these two forms of biparental care are visually represented on the tree in Figure 5A. Our data shows that out of 31 biparental species, 17 species in our dataset conform to the definition of complementary care. For example, in some Dendrobatid species, there is uniparental male attendance of terrestrial eggs, then the female alone transports the tadpoles to water filled plant cavities where she lays eggs for tadpoles to feed on over repeated visits. Dendrobatids are generally territorial and one species has been described as monogamous; however, information on mating system and territorially is very scarce to test the hypothesis proposed by the reviewer at a large scale. We now provide additional information in the revised ms on the natural history of biparental care that should help clarify the nature of complementary and overlapping care in amphibians (P21.L14-18).*

Page 16, Lines 330 – 333: You state that data from reviews were supplemented with data from the primary and secondary literature but don't supply any information on how (I also could not find this in the supp info). What searches were done to complement those studies, what were the search terms used, what was the depth of literature searched?

Response 27. We have assembled the largest and most detailed dataset of amphibian parental care to date by combining and cross-checking data from detailed reviews and secondary sources, supplemented with data from 401 primary sources. After scoring species based on information in reviews, large scale studies and their sources, and primary sources we had already had, we have searched for information on parental care for species that were ambiguous or that were classified differently across studies (please see Methods, Data collection, P19.L9-P20.L1; SI P1.L6-23). Specifically, to resolve ambiguity in the scoring of some species, we ran searches in Google Scholar and Web of Science, using the species' Latin name, alone and in conjunction with relevant keywords such as 'parental care', 'reproduction', 'mating', and 'life history', as described in our Methods section (P19.L9-P20.L1). We now provide this information in Supplementary Methods (SI P1.L6-23) where we provide the full list of 401 sources that we consulted to assemble the dataset (please see Supplementary References). Our dataset has been deposited in Dryad and will be available upon publication.

Supp Info, Page 1, Lines 17 – 21: You state that you did not include short term egg attendance (egg attendance immediately after birth) but what about egg attendance that did not go throughout the entire period. This is important because egg attendance for a prolonged period at the end of the developmental period is likely to exert a much more significant effect on the likelihood of post hatching care than egg attendance for a prolonged period of time at the begging. What data is there out there on duration of attendance and when that attendance occurs?

Response 28. We agree that this is an important point and one we have thought about extensively. In species with egg attendance, parents have been observed to guard the eggs, actively defend them against predators or conspecifics, protect them against pathogens, aerate them when developing in oxygen poor water pools and hydrate terrestrial developing eggs to avoid desiccation – these behaviours make egg attendance more costly than the reported cases of very brief post-hatching attendance and hydration reported in the glassfrogs (e.g. Delia et al 2017 J Evol Biol). Thus, we consider the latter cases as part of oviposition behaviour of the species rather than cases of parental care.

Parental care is often described qualitatively in natural history reports or as background information on the species in studies on other topics, and only a very small minority of studies provide quantitative data on care duration. There is qualitative information as parents have been frequently observed to attend the eggs from egg deposition to hatching into tadpoles or juveniles (e.g. oviparous caecilians Gomes et al 2012 South Am J Herpetology; frogs: Dugas et al. 2016 Behav Ecol; Townsend et al 1984 Animal Behaviour, among others). However, because the duration of egg attendance as quantitative measure is unknown for the vast majority of species, there is currently insufficient data for a large scale analysis of this variable. To address the reviewer's comment on egg attendance, we now provide more details in our description of this behaviour that should clarify how it differs from the brief attendance reported in the glassfrogs (Supplementary Information P2.L2-6).

Reviewer #3 (Remarks to the Author):

This manuscript describes broad-scale analyses of parental care behaviours in Amphibians. The breadth of the study is impressive and represents probably the most extensive data set of parental care for this taxon – this is particularly impressive given the wide range of care behaviours found in Amphibians. The analyses assess rates and patterns of transitions among care states and categories and show that simpler behaviours, such as egg attendance, evolve more rapidly than more complex care traits, such as viviparity. The analyses also show that biparental care is likely to be an unstable state, with transitions away from biparental care arising at a much higher rate than transitions towards biparental care. The manuscript

addresses questions of broad interest in a system that is particularly important for understanding the evolution of parental care. I do have some queries and suggestions but I do not anticipate that substantial new analyses are necessary (but see suggestions in point 2 and potentially point 4 below).

Thank you for the positive and constructive comments on the manuscript.

1) Viviparity as a subdivision of care does not seem well justified. I would argue that it is a life history trait that can result in different forms of offspring demands and alters the selection pressures on care behaviours, rather than being a care behaviour itself. This distinction might be helpful in terms of explaining differences in gains/losses. Perhaps behavioural traits are more labile than life history or physiological traits because changes in traits such as viviparity may require more complex developmental/genetic changes. Perhaps a more useful/justifiable division would be behavioural versus physiological (or anatomical) transitions (see lines 15-18 on page 4).

*Response 29. Viviparity and other morphological/anatomical traits are frequently considered a form of parental care; we thus follow the extensive literature on parental care and consider viviparity as a form of care (e.g. Royle et al. Eds 2012 *The evolution of parental care*, Oxford University Press; Clutton-Brock 1991 *The evolution of parental care*, Princeton University Press). We did indeed consider whether behavioural vs anatomical/physiological would be a suitable distinction in relation to complexity. However, amphibians exhibit such a diversity of care forms that this is not a straightforward distinction. For example, is brooding in the vocal sac or in the stomach a behaviour or an anatomical adaptation? Do we consider brooding on the back, where the eggs are not covered by parental skin, a behaviour, and if so, is it different from brooding in the vocal sac or the stomach? These questions are not trivial as, presumably, brooding in the vocal sac or the stomach requires some important physiological changes in the parent and offers opportunities to the developing embryos to manipulate parental resources that egg attendance cannot, but, perhaps not as extensively as viviparity. Unfortunately, we have too limited information to answer these questions with confidence as these adaptations are still very little studied, and we believe our study will stimulate important research on these forms of care. Following comments from the reviewers, we have now revised the text on the sections on trait complexity in the Introduction and added a new paragraph (P6.L11-P7.L2); please also see response 1 on a similar comment by Reviewer 1.*

2) Biparental care is very rare (17 species with complementary care and 14 with overlapping care) and I wonder whether this is problematic for the models. It is hard to tell from the figure, but it is not clear that any losses are actually observed so it is not clear where the inference of high rates of loss comes from and I was left wondering if the result is reliable or whether the model assumptions have been violated. With the inferred rate of loss being so much higher than the rate of gain of biparental care, it seems remarkable that biparental is observed at all.

The challenge of inferring gains and losses of binary traits have been discussed and in particular the potential importance of differential speciation and extinction has been highlighted (e.g. Goldberg, E.E. and B. Igić. 2008. On phylogenetic tests of irreversible evolution. *Evolution* 62:2727-2741; Goldberg, E.E., J.R. Kohn, R. Lande, K.A. Robertson, S.A. Smith, B. Igić. 2010. Species selection maintains self-incompatibility. *Science* 330:493-495). Perhaps biparental confers different speciation or extinction rates compared to uniparental care and this could explain the distribution of care. It may not be necessary to re-analyse the data using such models but at the least some consideration of these alternative mechanisms is needed.

Although plausible, I don't think that the results are convincing in terms of stability of biparental care, as opposed to care simply being rare. Perhaps a relatively straightforward way to assess the reliability of the results would be to simulate a multistate character evolving on the tree under the estimated transition rate parameters and assessing the simulated frequency of character states compared to the observed – this could be achieved with the `sim.character` function in the R package `diversitree`.

Response 30. We have taken these concerns very seriously and we thank the reviewer for suggesting the R package and function to evaluate the robustness of our results. We have carried out the suggested

simulations with the *sim.character* function in *diversitree* and we confirm that, using the transition rates estimated by *BayesTraits*, our results are robust. We simulated the evolution of a multistate discrete character (type of biparental care, 3 states: none, complementary, overlapping) evolving along the amphibian tree with the mean transition rates derived from the posterior distribution of the RJ Multistate analysis of the empirical data in *BayesTraits*. We repeated the simulations 100 times and derived a distribution of estimated number of species in each care state. The figure below shows that the empirical number of species with overlapping or complementary care (vertical red line) falls well within the simulated numbers.

Moreover, when plotted on the tree, the distribution of biparental care types at the tips based on the simulation mean approximates closely the scattered distributions of the empirical data across the tree (see figure below).

Our RJ Multistate analysis shows that both forms of biparental care evolve very slowly and are lost very rapidly (Figure 5 in the revised ms); indeed, if this was not the case and biparental care types were lost more slowly, we would observe many more closely related species sharing the same type of biparental care on the tips. We rely on the RJ Multistate models in BayesTraits for the analysis and interpretation of the results as these are the most advanced methods available for studies on the evolution of discrete variables (Currie & Meade (2014) in *Modern phylogenetic comparative methods and their application in evolutionary biology*, Springer); please also see our earlier Response 13. Crucially, by allowing for a reduction in overparametrisation and model complexity, RJ provides more reliable parameter estimates - this is very useful in cases where sample sizes from which to estimate parameters are not high and so a major strength of our analyses. Please note that the option of running analyses using RJ is not available with other methods. We note that we also ran the RJ Multistate analyses in triplicate and these converge on qualitatively very similar solutions (P22.L19-20) further confirming our results are robust.

With regard to speciation and extinction, we are not aware (and could not find in the literature) any a-priori hypothesis as to how and why parental care or biparental care or type of biparental care (with or without division of labour) should affect speciation or extinction rates. Thus, there is currently no identified mechanism by which parental care should affect speciation or extinction. We agree with the reviewer that reanalysing our data by incorporating speciation and extinction is not necessary; moreover, it would require the estimation of additional parameters (in conjunction with transition rates) which in turn will require an even larger sample of species with biparental care. Because there is no theoretical reason linking speciation and extinction to parental care, we do not discuss this in our ms.

In conclusion, the suggested simulation is highly concordant with the number of biparental care species we observe in the empirical data and in their scattered distribution across the tree; the RJ procedure in our analyses is extremely useful for estimating parameters when sample sizes are not high and the 3 independent runs converge on qualitatively very similar solutions indicating our results are robust. We believe that our results on the instability of biparental care are of great interest to many researchers in the field of parental care, reproductive and life history strategies. Likewise, the way we have classified and investigated the stability of biparental care forms with and without division of labour will provide a useful template to others who wish to address this question in other taxa or are interested in studying the evolution of other diverse and complex traits.

In addition, it would be helpful to see in more detail the distribution of biparental care on the tree. These details are in figure 4, panel a but it is hard to read. It either needs to be much bigger, or perhaps replotted to show more clearly where overlapping and complementary biparental care evolves (i.e. don't plot the grey circles and instead state that nodes lacking a symbol have no biparental care).

Response 31. We acknowledge that the figure at first submission had low resolution. Because few species have biparental care, are scattered across a large tree, it is difficult to visualize them easily in a figure. We have tried the solutions proposed by the reviewer but we found that the figure was even less clear, mostly because of the sparse distribution of biparental care types on the tree. Our major objective of Figure 5A is to show the distribution of the data on the tips of the phylogeny (please see response 4 in our use of stochastic character mapping for visualisation purposes rather than as analytical tool). To try and help better see the species on the tree we have increased the size and especially the resolution of Figure 5A so that readers can zoom in and identify species with the different types of care. With this submission, we provide high resolution files for all panels of figures with the phylogeny.

3) The raw data do not seem to be available – will they be deposited to e.g. Dryad on publication?

Response 32. Yes, the dataset has been deposited in Dryad and will be available at publication. We have now added an explicit statement of data availability in the revised ms; DOI to be added on publication (P20.L6-8).

4) Losses in the table in figure 1 don't seem to be consistent with the transition rates, most obviously the rate of loss for tadpole feeding is clearly non-zero in panel c but the number of losses is estimated to be zero. The patterns for gains seem consistent (high rate associated with high numbers of gains), but losses don't appear to tally at all. I may have missed something but I wonder if there is an error in reporting of these results.

Response 33. *please see response 13 to a similar point raised by reviewer 2. Briefly, BayesTraits has no graphic tool associated with it, thus we use stochastic character mapping just for visualisation purposes and to provide readers less familiar with transitions rates a number of gain and losses. The estimated number of gains and losses using stochastic character mapping is however one realisation of all possible models that RJ Multistate analyses can identify. Moreover, the MCMC analysis in BayesTraits is more powerful and also allows for the use of the RJ procedure (please see response 30). While for all other traits in Figure 1, stochastic character mapping provides a reasonably close match to the results of the RJ Multistate analysis, it does not for tadpole feeding. We do agree that this has created a lot of confusion. To resolve the issue and clarify what results we rely on when we draw conclusions, we have now deleted the table of gains and losses as this is based on stochastic character mapping (Figure 1D of the previous submission) and the Supplementary Figure 1. We now provide instead additional details to help readers less familiar with the interpretation of transition rates (P6.L7-10; P9.L12-16).*

Additional minor comments:

It would be helpful to report a range or HPDs for transition rates and number of gains losses e.g. page 8, line 2, and in the supplementary tables (a range would be more helpful than reporting mean, median and mode).

Response 34. *We now include the 95% HPD interval for all estimated transition rates in the supplementary tables for all BayesTraits analyses as suggested. Please note that we also show the posterior distributions of all parameters for all analyses in the relevant figures in two different ways; as box plots for easier comparison across parameters and as density. We still report in the supplementary tables the mean, mode and median together with the % of models estimating a parameter as equal to 0 because the posterior distributions of parameter estimates from RJ analysis may not be bell-shaped, and therefore the 95% HPD may not be useful to describe these distributions. Indeed, the mean is useful to represent the average parameter estimate and we report this in the summary diagrams in the figures; the median matches the box—plots in the panels showing the posterior distributions, while the mode helps identify the most common value in the posterior distribution. These, together with the % of models estimating a transition rate to be equal to 0 and the figures, provide a detailed, comprehensive and informative description of the posterior distribution from RJ models.*

The estimated number of gains and losses can currently only be derived from stochastic character mapping summaries as BayesTraits only provides estimates of transition rates (please see our response 13 and 33). Please also note that estimating the gains and losses with stochastic character mapping is comparable to fitting a multistate model, which is not equivalent to the Discrete framework that we used for more than half of the analyses. Considering these issues, and the fact that our presentation of number of gains and losses has created much confusion being derived from a different method, we no longer report the number of gains and losses for any analysis.

Page 3 line 6: “Not only does parental care increase the fitness of offspring and parents”: Is this true? If care increases fitness of parent and offspring then there should not be intrafamilial conflict. Isn't the point that care can effect fitness of individuals (offspring, mother, father) differentially and it is this variation that cause conflict?

Response 35. *Good point, we revised this sentence and changed ‘increase’ with ‘affect’ (P4.L6).*

Page 5, lines 10-12: This is a quite hard to follow (what does “parental care forms of diverse complexity”

mean?).

This sentence has been deleted during revision.

Methods, page 17, line 14. The description of tree scaling is ambiguous – how was the tree scaled and why?

Response 36. *As recommended in the BayesTraits manual (see below) and as confirmed by the developers of BayesTraits (pers. comm.), scaling the tree by a constant allows the algorithm to better explore parameter space and better estimate transition rates, particularly when transition rates are expected to be very small (as in our case). Note, however, that this does not alter conclusions as both the parameters and the branches are scaled by the same constant. In our analyses we scaled the tree using the default settings, i.e. to have a mean of 0.1. We now provide information on how and why we scaled the tree in the revised ms (P21.L22-P22.L5).*

From the BayesTraits V3 manual: “Model parameters are dependent on branch lengths, as branch lengths can come in differing units, years, millions of years, expected number of substitutions, it is recommended that the branch lengths are scaled to have a mean of 0.1 for MultiState and Discrete models, as this prevents the rates becoming small, hard to estimate or search for.”

Also, why use this rather than the more recent Pyron and Jetz amphibian tree?

Response 37. *We decided to use the Pyron (2014) amphibian timetree which at present is the most comprehensive tree (3,309 extant species) constructed exclusively from molecular data. Conversely, the more recent Pyron and Jetz’s (2018) amphibian tree is an extension of Pyron (2014) including 7,238 amphibian species, but it is assembled using a combination of molecular inference and taxonomic assignment, so that the phylogenetic position of over 3000 species is based on taxonomy. For our study, however, we want to use a molecular tree to avoid adding further uncertainty and potential biases.*

Figure 1 - Is grey absence or lack of data? The figure is hard to read because of the small size.

Response 38. *We have no missing data in the dataset for this study (P9.L6-10; P20.L6-8; Table S1 in SI) and across all figures ‘grey’ indicates the absence of a trait. To address the reviewer’s point, we now make this clearer by adding this information also in the legend of all relevant figures. To facilitate reading this figure, we have also enlarged Figure 1A and provide a high resolution image of it that allow zooming accurately.*

Reviewers' Comments:

Reviewer #1:

Remarks to the Author:

The authors have adequately addressed my previous comments in their revision.

Reviewer #2:

Remarks to the Author:

I was a previous reviewer of this MS and the authors have done a great job of addressing my previous concerns regarding the MS and adding some additional focus and content to really emphasise the novel components of the MS. I do have a number of minor concerns still regarding this MS which I detail below. If they can be addressed then I believe that this MS will make a valuable contribution to our understanding of how parental care evolves.

1. One thing that I think might help the interpretability of the MS overall is if there was a diagram indicating how the different care/developmental periods overlap. That is eggs, viviparity, brooding with direct development, tadpoles, juveniles, ect - perhaps indicating the number of species with each form of developmental trajectory. This would really help the reader interpret the sequence of caring periods for each developmental sequence and thus visualize how care types at early stages of development actually correspond with later development for each of the different developmental trajectories observed in this group. Of course this, may not be possible if the diversity of developmental trajectories that occur across amphibians is extremely high - but to some degree it must not be if you want to make generalizations across these different forms of development.

Page 11, Lines 3 - 5: I am not convinced that this is a particularly unexpected result - I would expect that egg attendance is the easiest trait to evolve - because parents are most likely to be present at the production of eggs - but less likely to be present at the hatching of those eggs. Thus, attendance of the eggs simply requires parents to hang around immediately after their production - but attendance at the tadpole and juvenile stage requires prolonged association (either via initial egg attendance - or via other processes like viviparity - see Halliwell et al., 2017 for similar arguments). In this context you don't necessarily need an explanation that invokes differential selective pressures (lines 8 - 9) - it's all about opportunity (which is kind of inferred on lines 10 - 14).

Page 14, Lines 6 - 8: I found it interesting that egg brooding and viviparity were did not facilitate prolonged care post birth/brooding. It seems like these traits would pose an as likely scenario as egg attendance for facilitating prolonged care. I wonder if this should be discussed a little more.

Page 14, Lines 12 - 15: Minor point but following on from a point above - I am not sure that it immediately follows that parent-offspring conflict and sibling competition facilitate increases in care complexity - instead the reduction of conflict and the increase in competition is likely to stabilize the associations which then set the stage for the further elaboration of care. Fundamental to this, to my interpretation, would simply be the presence of offspring and parents together in some form and that itself provides a novel selective environment for promoting either even more prolonged associations (e.g., care at other stages of development) or for the evolution of more complex forms of care within that developmental stage (e.g., via the emergence of novel behaviours or the co-option of behaviours that function in novel social scenarios). This will of course only occur if the mediation of conflict and cooperation allows associations to be stabilized - but that conflict and cooperation does not necessarily lead to the emergence of those extended forms of parental care per se.

Page 16, lines 15 - 23: How much actual data is there available to actually study transitions between uni and bi-parental care at the level of individual care traits? From the the next page it appears as if there are only 31 species that exhibit bi-parental care - does that really supply the requisite diversity across specific care types to allow for examining transitions for specific care types (e.g., tadpole transport). I realise that you note that these analyses are only undertaken on traits for which you have sufficiently large sample sizes but I was still not clear whether we can be confident in drawing any strong conclusions from trait specific examinations of biparental care.

Page 18, lines 7 - 23: Following on from the above I thought some of your final conclusions are slightly overstated given what your data suggested. Specifically, your data did not allow you to infer anything about relative costs and benefits of different traits nor anything about the extent of parent vs offspring control. While these things can be discussed in the main body of the discussion, with additional suggestions with respect to moving forward with research to test these ideas, I was not convinced they should be coaxed as major conclusions per se. Furthermore, could some of these ideas be tested, particularly the latter idea, by examining whether the evolution of parental care has resulted in a change in offspring phenotype in the direction expected if parental and offspring traits were co-evolving? Do offspring of species with strong prolonged care show evidence of shifts towards more altricial offspring?

Reviewer #3:

Remarks to the Author:

I appreciate the author's responses and revisions to the manuscript. This is a very nice paper and I have just a couple of minor text edits to suggest.

line 16: "...are more common because they are easier to evolve;"

page 21, line 22: Clarify that scaling is to a mean branch length - the current wording is ambiguous since rescaling is usually to a set tree height (root to tip distance)

Signed

Gavin Thomas

REVIEWERS' COMMENTS:

Reviewer #1 (Remarks to the Author):

The authors have adequately addressed my previous comments in their revision.

Reviewer #2 (Remarks to the Author):

I was a previous reviewer of this MS and the authors have done a great job of addressing my previous concerns regarding the MS and adding some additional focus and content to really emphasise the novel components of the MS. I do have a number of minor concerns still regarding this MS which I detail below. If they can be addressed then I believe that this MS will make a valuable contribution to our understanding of how parental care evolves.

1. One thing that I think might help the interpretability of the MS overall is if there was a diagram indicating how the different care/developmental periods overlap. That is eggs, viviparity, brooding with direct development, tadpoles, juveniles, ect - perhaps indicating the number of species with each form of developmental trajectory. This would really help the reader interpret the sequence of caring periods for each developmental sequence and thus visualize how care types at early stages of development actually correspond with later development for each of the different developmental trajectories observed in this group. Of course this, may not be possible if the diversity of developmental trajectories that occur across amphibians is extremely high - but to some degree it must not be if you want to make generalizations across these different forms of development.

Thanks for this valuable suggestion; we now include such a diagram as Supplementary Figure 1. Please note the sample sizes are also available in Supplementary Table 1.

Page 11, Lines 3 - 5: I am not convinced that this is a particularly unexpected result - I would expect that egg attendance is the easiest trait to evolve - because parents are most likely to be present at the production of eggs - but less likely to be present at the hatching of those eggs. Thus, attendance of the eggs simply requires parents to hang around immediately after their production - but attendance at the tadpole and juvenile stage requires prolonged association (either via initial egg attendance - or via other processes like viviparity - see Halliwell et al., 2017 for similar arguments). In this context you don't necessarily need an explanation that invokes differential selective pressures (lines 8 - 9) - it's all about opportunity (which is kind of inferred on lines 10 - 14).

We agree with the reviewer that attendance at the later stages (tadpole and juvenile) is also a matter of opportunity as well as function, and our first analysis by care form (Page 9, Line 16-Page 10, Line 2) can help address this. The potential different opportunities for evolving a care function (such as attendance) at different stages may also affect the rate of gain and loss and thereby mask differences in evolutionary rates due to complexity. Our analysis by function addresses this as it allows us to remove the potential confounding effect of opportunity. In support of this, we have reviewed the literature which unambiguously suggests that function (irrespective of opportunity) is indeed the same across stages in Amphibians, including for attendance (P10.L7-12). The combined results of these two sets of analyses therefore provide a fuller and more comprehensive picture of the evolution of care forms than either can achieve on its own. The sentence highlighted by the reviewer aims to point out that tadpole attendance evolves at a low rate comparable to that of complex traits like viviparity. In the following sentences (P11.L9-17), we explicitly discuss the role of differential opportunity in determining the likelihood for care to evolve at different stages. To address the reviewer's comments, we have added further clarification on the rationale of our analysis by function (P10.L4-7) and slightly revised the sentence flagged by the reviewer to emphasize the comparison between the rates of tadpole attendance and complex traits (more than the comparison between tadpole attendance and egg attendance; P11.6-9).

Page 14, Lines 6 - 8: I found it interesting that egg brooding and viviparity were did not facilitate prolonged care post birth/brooding. It seems like these traits would pose an as likely scenario as egg attendance for facilitating prolonged care. I wonder if this should be discussed a little more.

We agree that the hypothesis that egg brooding with direct development and viviparity should promote the evolution of care at the juvenile stage is sensible; however, our results show clearly that the hypothesis is not supported in Amphibians. Because the ms is already complex, we think it is not necessary to discuss in more details non-significant results of complementary analyses.

Page 14, Lines 12 - 15: Minor point but following on from a point above - I am not sure that it immediately follows that parent-offspring conflict and sibling competition facilitate increases in care complexity - instead the reduction of conflict and the increase in competition is likely to stabilize the associations which then set the stage for the further elaboration of care. Fundamental to this, to my interpretation, would simply be the presence of offspring and parents together in some form and that itself provides a novel selective environment for promoting either even more prolonged associations (e.g., care at other stages of development) or for the evolution of more complex forms of care within that developmental stage (e.g., via the emergence of novel behaviours or the co-option of behaviours that function in novel social scenarios). This will of course only occur if the mediation of conflict and cooperation allows associations to be stabilized - but that conflict and cooperation does not necessarily lead to the emergence of those extended forms of parental care per se.

We agree with the reviewer that the close associations between the parents and offspring offers opportunity for care to evolve. Our point, though, is that the increase vs decrease of care duration depends on which care form evolves and who is in control of resource allocation (the parent or the offspring). Our suggestion therefore goes beyond what is proposed by the reviewer, in that we emphasize how care duration changes depending on care form. We also agree that conflict will not always lead to extended care or complex care in every case. However, our aim here is to identify which potential driver may underpin the evolution of prolonged care and more elaborate care, and conflict is a strong candidate for that. In other words, we identify a potential strong mechanism underpinning 'the selective environment' that the reviewer mentions. Thus, we do not think there is any contradiction between what the reviewer states and what we write in our discussion of this set of results.

Page 16, lines 15 - 23: How much actual data is there available to actually study transitions between uni and bi-parental care at the level of individual care traits? From the the next page it appears as if there are only 31 species that exhibit bi-parental care - does that really supply the requisite diversity across specific care types to allow for examining transitions for specific care types (e.g., tadpole transport). I realise that you note that these analyses are only undertaken on traits for which you have sufficiently large sample sizes but I was still not clear whether we can be confident in drawing any strong conclusions from trait specific examinations of biparental care.

The sample sizes for all analyses, including those referred to in these lines, are reported in the caption of the Supplementary Tables. For the analysis of egg attendance, the sample sizes of each of the four states, indicated in Supplementary Table 6b, are as follows: 17 species are biparental, 164 male uniparental, 148 female uniparental, and 993 lack egg attendance. For the analysis of Tadpole transport, the sample sizes of each of the four states, indicated in Supplementary Table 6c, are as follows: 3 species are biparental, 51 are male uniparental, 9 female uniparental and 1259 lack tadpole transport.

As we note in the ms (P13.L17-20), we only performed Discrete analyses for care behaviours (egg attendance and tadpole transport) with sufficiently large sample sizes. Moreover, we further confirm that our results are robust as the RJ procedure we used is designed to accommodate small sample sizes by

reducing model complexity, all parameters have ESS of 1000 or more (see Supplementary Tables 2-7), their traces show adequate mixing and convergence, and three replicate runs of each analysis converged on qualitatively similar solutions (P22.L18-P23.L2; P23.L7-11); please see also our detailed response to similar comments in the previous round of review). To address the reviewer comments, we add further clarification on the RJ procedure (P23.L1-2).

Page 18, lines 7 - 23: Following on from the above I thought some of your final conclusions are slightly overstated given what your data suggested. Specifically, your data did not allow you to infer anything about relative costs and benefits of different traits nor anything about the extent of parent vs offspring control. While these things can be discussed in the main body of the discussion, with additional suggestions with respect to moving forward with research to test these ideas, I was not convinced they should be coaxed as major conclusions per se. Furthermore, could some of these ideas be tested, particularly the latter idea, by examining whether the evolution of parental care has resulted in a change in offspring phenotype in the direction expected if parental and offspring traits were co-evolving? Do offspring of species with strong prolonged care show evidence of shifts towards more altricial offspring?

We agree with the reviewer and have deleted the reference to costs and benefits from the final paragraph of the Discussion. Given the length of the paper, we refrain from adding further discussion on points related to offspring phenotypes. As we noted in our detailed response in the previous round of review, there is limited quantitative data to comment on the degree of altriciality in relation to care in amphibians.

Reviewer #3 (Remarks to the Author):

I appreciate the author's responses and revisions to the manuscript. This is a very nice paper and I have just a couple of minor text edits to suggest.

We thank the reviewer for the positive comments on our ms.

line 16: "...are more common because they are easier to evolve;"

Changed as suggested (P4.L16).

page 21, line 22: Clarify that scaling is to a mean branch length - the current wording is ambiguous since rescaling is usually to a set tree height (root to tip distance)

Thanks for the suggestion; we changed the sentence as suggested (P22.L14).

Signed
Gavin Thomas